



**Evaluating the Consistency and Continuity of Pixel-Scale Cloud**
**Property Data Records From *Aqua* and *SNPP***
Qing Yue[1], Eric J. Fetzer[1], Likun Wang[2], Brian H. Kahn[1], Nadia Smith[3], John Blaisdell[4], Kerry
G. Meyer[5], Mathias Schreier[1], Bjorn Lambrigtsen[1], and Irina Tkatcheva[1]
[1]Jet Propulsion Laboratory, California Institute of Technology, Pasadena, CA
[2] Earth System Science Interdisciplinary Center, University of Maryland, 5825 University Research Court, Suite 4001,
College Park, MD 20740.
[3]Science and Technology Corporation, 10015 Old Columbia Road, Columbia, MD 21046
[4]Science Applications International Corporation, 12010 Sunset Hills Road, Reston, VA 20190
[5]NASA Goddard Space Flight Center, Greenbelt, MD.

Correspondence to: Qing Yue (qing.yue@jpl.nasa.gov)
**Abstract**
The *Aqua*, *SNPP*, and *JPSS* satellites carry a combination of hyperspectral infrared sounders
(AIRS, CrIS) and high-spatial-resolution narrowband imagers (MODIS, VIIRS). They provide an
opportunity to acquire high-quality long-term cloud data records and are a key component of the
existing Program of Record of cloud observations. By matching observations from sounders and
imagers across different platforms at pixel scale, this study evaluates the self-consistency and
continuity of cloud retrievals from *Aqua* and *SNPP* by multiple algorithms, including the AIRS
Version-7 retrieval algorithm and the Community Long-term Infrared Microwave Combined
Atmospheric Product System (CLIMCAPS) Version-2 for sounders, and the Standard *Aqua*-
MODIS Collection-6.1 and the NASA MODIS-VIIRS continuity cloud products for imagers.
Metrics describing detailed statistical distributions at sounder field of view (FOV) and the joint
histograms of cloud properties are evaluated. These products are found highly consistent despite
their retrieval from different sensors using different algorithms. Differences between the two
sounder cloud products are mainly due to cloud clearing and treatment of clouds in scenes with
unsuccessful atmospheric profile retrievals. The sounder subpixel cloud heterogeneity evaluated
using the standard deviation of imager retrievals at sounder FOV shows good agreement between
the standard and continuity products from different satellites. However, impact of algorithm and
instrument differences between MODIS and VIIRS is revealed in cloud top pressure retrievals and
in the imager cloud distribution skewness. Our study presents a unique aspect to examine NASA's
progress toward building a continuous cloud data record with sufficient quality to investigate
clouds' role in global environmental change.



## 1. Introduction

Clouds play an important role in Earth's energy balance and hydrological cycle. They occur with processes involving atmospheric radiation, thermodynamics, and dynamics at various spatial and temporal scales, making clouds a crucial component of the weather and climate system. With daily regional and global coverage, space observations provide a unique vantage point to monitor the change of the cloud properties in the climate system across different time scales. This offers an important observational basis to resolve cloud processes in the background atmospheric circulation, which is widely recognized as a critical challenge within Earth Sciences (Bony et al. 2015, IPCC 2013). The 2017 US National Academy Decadal Survey (ESAS 2017) has noted the importance of long-term and sustained observations of many key components of the Earth system, including continuity measurements of clouds. Many of these observations are obtained from the existing Program of Record (POR). Since the "POR forms the foundation upon which the committee's recommendations are established" (ESAS 2017), it is crucial to evaluate whether a self-consistent and continuous POR for cloud-related variables is indeed available with sufficient data quality and spatio-temporal coverage.

Cloud retrievals from the NASA's Earth Observing System (EOS) satellites, including *Terra* and *Aqua*, the joint NASA/NOAA Suomi National Polar-orbiting Partnership (*SNPP*), and NOAA's new generation of Joint Polar Satellite System (*JPSS*) series weather satellites, are a key component in the POR for cloud properties. Through efforts on continuity and consistency by rigorous instrument mission design and ongoing algorithm development, these satellites provide high quality, long-term cloud data records derived from the Top of Atmosphere (TOA) radiances observed across a wide range of the emission and reflection spectrum. Particularly, *Aqua*, *SNPP*, and *JPSS-1* (now *NOAA-20*), which were launched in 2002, 2011, and 2016, respectively, carry



high spatial resolution narrowband imagers, hyperspectral infrared (IR) sounders, and microwave
(MW) sounding measurements. As a result, observations with similar spatial resolution and
coverage, and similar spectral resolution at analogous wavelengths are obtained from different
satellites. For *Aqua*, this instrument trio consists of the Atmospheric Infrared Sounder (AIRS), the
Advanced Microwave Sounding Unit (AMSU), and the Moderate Resolution Imaging
Spectroradiometer (MODIS). For *SNPP* and *JPSS*, the trio includes the Cross-track Infrared
Sounder (CrIS), the Advanced Technology Microwave Sounder (ATMS), and the Visible Infrared
Imaging Radiometer Suite (VIIRS).

Retrieval algorithms to maintain the continuity of the data records across these platforms have

been developed. For joint retrievals by IR and MW sounders such as AIRS/AMSU and
CrIS/ATMS, the Community Long-term Infrared Microwave Combined Atmospheric Product
System (CLIMCAPS; Smith and Barnet, 2019) provides cloud properties together with vertical
profiles of atmospheric temperature, water vapor, and trace gases, as well as surface conditions.
For imagers like MODIS and VIIRS, the NASA MODIS-VIIRS continuity cloud products have
been developed for both cloud mask (CLDMSK; Frey et al. 2020) and cloud optical properties
(CLDPROP; Platnick et al. 2021). These continuity algorithms have heritage with NASA
operational retrieval products previously developed for individual sensors and satellites, such as
the AIRS Science Team retrieval algorithm Version 7 (AIRS V7, Yue and Lambrigsten 2017, 2020)
in the case of CLIMCAPS, and the Standard *Terra/Aqua* MODIS Collection 6.1 cloud retrievals
(MOD35/MYD35, MOD06/MYD06; Baum et al. 2012, Platnick et al. 2017) in the case of
MODIS-VIIRS. However, significant differences exist between the standard and continuity
algorithms, as the focus of the continuity algorithms is to minimize the impact of instrument
between platforms.





The sounder-imager combination on the same sun-synchronous polar-orbiting satellite,
together with the temporal coverage overlap between satellites, provides opportunities to utilizing
spectral and spatial capabilities from different sensors at global scale. Previous studies have shown
the benefits of using the combined information to intercalibrate and test radiometric consistency
among sensors (Tobin et al. 2006, Schreier et al. 2010, Wong et al. 2015, Gong et al. 2018); cross-
validate the retrievals (Nasiri et al. 2011, Kahn et al. 2014); further improve atmospheric and
surface geophysical parameter retrievals (Irion et al. 2018, Yao et al. 2015); provide simultaneous
observations to resolve complex physical processes (Yue et al. 2013, 2016, 2019, McCoy et al.
2017); quantify the subpixel heterogeneity (Li et al. 2004, Kahn et al. 2015); and enhance the
utilization of satellite observations in numerical weather prediction and climate models (Eresmaa
2014). Therefore, the sounder-imager combination is an important aspect of data record continuity
and consistency among sensors across different platforms. This helps provide robust monitoring
of long-term changes in cloud properties, an important capability expected from the POR.
Pixel-scale analyses are an effective and unique way to investigate the consistency and
continuity of these data records because of the one-to-one relationships established by these
comparisons and their direct links to algorithm performance. This includes examining differences
of (1) the same physical parameters observed by different sensors or satellites but processed using
the same (or similar) algorithms, and (2) the same parameters obtained from the same sensor but
from different algorithms. Both of these differences are quantified at the pixel scale in this study.
The cloud properties determined by the sounder and imager pairs on board *Aqua* and *SNPP*,
namely AIRS/MODIS and CrIS/VIIRS, are investigated using the collocated sounder-imager
fields of view (FOVs) for sets of pixels obtained during Simultaneous Nadir Observations (SNOs)
between *Aqua*-AIRS and *SNPP*-CrIS. This approach ensures nearly identical viewing geometry



by the two satellites while pixel-scale cloud assessment is carried out by comparing cloud
parameters determined by hyperspectral IR sounders and high spatial resolution imagers at the
minimum spatial scale of individual instrument fields of view. Using this approach, products from
both the heritage NASA standard retrieval algorithms and the newly-developed continuity cloud
algorithms are analyzed (Table 1). This is essential for retrieval algorithm development and cross-
validation of multiple sensors and products on *Aqua* and *SNPP*, and also important for data
continuity extending to future *JPSS* satellites.

**2.  Data and Methodology**
2.1 Cloud products and algorithms

Table 1 summarizes the cloud parameters analyzed in this study from various Level 2 (L2)

retrieval products derived from the sounders and imagers aboard *Aqua* and *SNPP*. For AIRS and
MODIS, both the standard operational and continuity products are evaluated: the AIRS V7 and
CLIMCAPS-*Aqua* Version 2 (V2) retrievals for AIRS, and the Collection 6.1 *Aqua* MODIS
Atmosphere Level 2 Cloud Product (MYD06) and Version 1.1 NASA *Aqua* MODIS Continuity
Cloud Property Products (CLDPROP_MODIS). For *SNPP*-CrIS and -VIIRS, only the continuity
products are evaluated, which are the V2 CLIMCAPS-*SNPP* and Version 1.1 *SNPP*-VIIRS
Continuity Cloud Property Products (CLDPROP_VIIRS). The CLIMCAPS-*SNPP* products were
produced using Version 2 of the CrIS Level-1B product in Nominal Spectral Resolution (NSR)
and Full Spectral Resolution (FSR), which differ in the spectral resolution of the shortwave and
mid-IR CrIS observations transmitted from *SNPP* (Monarrez et al. 2020). The spectral resolution
differences cause subtle differences between the CLIMCAPS FSR and NSR retrievals, especially
in the upper tropospheric humidity and trace gases (Wang et al. 2021).



In both the AIRS V7 and CLIMCAPS algorithms for AIRS and CrIS, the radiatively effective
cloud amount (effective cloud fraction, ECF) and cloud top pressure (CTP) are retrieved by
matching the calculated cloudy radiances with the observed radiances for a set of channels that are
sensitive to clouds. Then the cloud top temperature (CTT) is derived as the atmospheric
temperature matching the retrieved CTP. In this process, best estimates of surface and atmospheric
parameters are used to calculate the cloudy radiances, either from the *a priori* state or from the
physical retrieval after the cloud clearing step (Susskind et al. 2003, Susskind et al. 2006, Smith
and Barnet 2019). The cloud clearing approach (Chahine 1974) is applied in both the AIRS Science
Team algorithms and CLIMCAPS. It predicts a single cloud cleared radiance at one AMSU or
ATMS field of regard (FOR) using *a priori* temperature, water vapor, and surface information and
a linear combination of IR radiances from nine AIRS or CrIS FOVs that are co-registered with one
AMSU or ATMS FOR (Susskind et al. 2003). The cloud cleared radiances are subsequently used
to retrieve surface and atmospheric parameters. Flowcharts of the retrieval steps and differences
in these two sounder retrieval systems are given in Thrastarson et al. (2021).
The ECF is the product of cloud areal fraction and the IR cloud emissivity, the latter of which
is assumed to be spectrally flat in the retrieval of ECF (Susskind et al. 2003). Previous studies
show that the AIRS ECF is consistent with the cloud properties such as the cloud frequency and
cloud optical depth measured by CloudSat and MODIS (Yue et al. 2011, Kahn et al. 2014). The
AIRS and CrIS retrievals of ECF and cloud top properties (CTT and CTP) are reported for up to
two cloud layers in each IR sounder FOV (~13.5 km spatial resolution at nadir).
There are distinct differences between the AIRS V7 and CLIMCAPS V2 algorithms regarding
cloud retrievals, summarized here. The first major difference is how cloud clearing is iterated in
the retrieval flow. The second major algorithm difference is quality control (QC) procedures when



1) the physical retrieval of atmosphere and surface is not successful, and 2) the final-stage cloud
clearing is not successful (Susskind et al. 2014). The third major difference is the choice of the
prior states for the two algorithms. The AIRS Science Team algorithms, including both V6 and
V7, iterate cloud clearing multiple times, and cloud parameters are determined after the last
iteration of cloud clearing using the retrieved surface and atmospheric conditions (Fetzer et al.
2020). In contrast, CLIMCAPS V2 performs a single cloud clearing pass and cloud properties are
retrieved using the surface and atmospheric parameters from successful retrievals of surface and
atmospheric properties (Smith and Barnet 2019, Thrastarson et al. 2021). The QC procedure used
in the two sounder cloud retrievals are also different. AIRS V7 produces case-by-case QC
indicators for each retrieved variable; while CLIMCAPS V2 derives one QC value based on the
cloud clearing and retrieval status of temperature and water vapor, and the same QC value is
assigned to all retrieved variables for the given FOV, including the cloud parameters. Particularly,
in AIRS V7 cloud retrieval process, the final stage of cloud clearing and cloud retrievals uses the
surface and atmospheric variable retrievals, except for cases over ocean when the retrieved surface
temperature differs from the first guess by more than 5 K. For these cases, the surface temperature
and surface emissivity from the *a priori* are used instead, and cloud properties retrieved under this
condition are flagged as valid with QC=1, indicating successful cloud retrievals but potentially
higher uncertainty than QC=0. This surface test effectively filters out cases when the cloud top is
misidentified as surface and causes extremely small ECF values for overcast cloudy conditions
over ocean.  For ~1% of cases the final cloud retrieval step does not complete successfully, and a
QC=2 flag is assigned to cloud parameters to indicate invalid retrievals. As a result, the AIRS V7
cloud retrievals produce a much higher percentage of cases with successful cloud retrievals (cloud
variable QC=0 or QC=1) than its temperature and water vapor profile products. For CLIMCAPS



V2, cloud clearing is not iterated and cloud parameters follow the QC procedure in the physical
atmospheric state retrievals. As a result, a much larger number of cases with QC=2 cloud retrievals
are reported by CLIMCAPS V2 compared to AIRS V7, especially for cloudier conditions or cases
with large cloud clearing errors, typically those FORs with low cloud contrast between associated
FOVs. Different *a priori* in the two retrieval systems impact their cloud retrievals. AIRS V7 uses
the Stochastic Cloud Clearing / Neural Network (SCCNN) solution as *a priori* on atmospheric
temperature and water vapor profiles and surface temperature trained using a few months of
European Center for Medium-Range Weather Forecasting (ECMWF) model analyses and
AIRS/AMSU radiances (Milstein and Blackwell 2016). For land and sea ice surface emissivity
prior estimates, AIRS V7 uses the University of Wisconsin – Madison Baseline Fit Emissivity
database (Seemann et al. 2008), which is based on the monthly climatology of MODIS land surface
emissivity product (MOD11) in 2008 (Thrastarson et al. 2021). The CLIMCAPS system (Smith
and Barnet 2020, Smith et al. 2021), instead, uses concurrent fields from the Version 2 Modern-
Era Retrospective analysis for Research and Application (MERRA-2, Gelaro et al. 2017) as the *a*
*priori* and implements the Combined ASTER (Advanced Spaceborne Thermal Emission and
Reflection Radiometer) and MODIS Emissivity database for land surface (Hook 2019). Over
ocean, both systems use the Masuda IR sea surface emissivity model (Masuda et al., 1988) as
modified by Wu and Smith (1997). Since the *a priori* temperature, water vapor, and surface
properties are used in the cloud clearing step, differences in the *a priori* contribute to the
differences between the retrieval products, including cloud properties (Yue and Lambrigtsen 2020,
Yue et al. 2021). Cloud clearing plays an important role in both retrieval systems, and physical
retrievals of surface and atmospheric parameters are obtained from the cloud cleared radiances,
which, in turn, impact the determination of cloud properties.



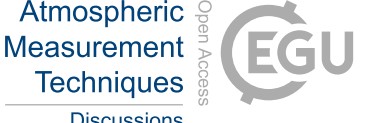
In addition to these major differences, the two sounder retrieval systems differ in the prior
estimates used for ECF and CTP. CLIMCAPS starts the cloud retrieval with background estimates
of 0.5 and 0.25 ECF at 350 hPa and 800 hPa CTP for the upper and lower cloud layers, respectively.
AIRS V7 uses 1/6 ECF at 350 hPa for the upper layer, and 1/3 ECF at 850 hPa (or 100 hPa above
surface in elevated terrain) for the lower cloud layer.  However, since the final cloud retrievals of
both systems are shown to diverge significantly from their prior (Yue and Lambrigtsen 2020, Yue
et al. 2021), it is unlikely that different cloud prior estimates are a main contributor to the sounder
cloud retrieval product differences.
Although their spectral resolution is coarser than that of AIRS and CrIS, instruments like
MODIS and VIIRS provide high spatial-resolution cloud properties through information in
multiple narrowband channels covering the visible and IR spectral regions. However, significant
differences exist between the two imagers. MODIS measures the reflectance or radiance in 36
spectral bands, while VIIRS has an analogous subset of these bands (20 channels) plus a day/night
visible channel (Oudrari et al. 2015). The lack of near-IR and IR water vapor and $CO_2$ absorption
channels in VIIRS has important implications on the available information content for clouds with
respect to MODIS. This impacts the determination of clouds, especially the detection of multi-
layer clouds and clear sky in polar night conditions, and the determination of cloud thermodynamic
phase. It also impacts the retrieval of cloud-top properties, especially for high thin clouds.
Moreover, the difference of spectral location of the VIIRS 2.25 µm channel compared to the
analogous 2.13 µm MODIS channel has implications on the retrievals of cloud particle size, optical
depth, and thermodynamic phase (Platnick et al. 2020). On the other hand, VIIRS provides a higher
spatial resolution of 750 m at nadir in cloud property retrievals, compared to the 1-km resolution
in the Collection 6.1 MYD06 and cloud mask products. In addition, VIIRS has an onboard detector

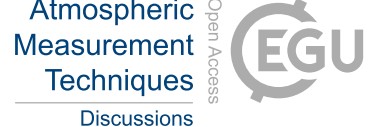

aggregation scheme that limits the across-swath pixel growth. VIIRS edge of scan pixel size is
roughly 1.625 km x 1.625 km versus roughly 2km x 4.9 km for MODIS (Platnick et al. 2021). The
MYD06 products have been shown to provide stable and well characterized cloud data records
since 2002 (e.g. Yue et al. 2017). Given these instrument differences between MODIS and VIIRS,
and a need to develop a continuous data record extending beyond the MODIS era, the MODIS-
VIIRS CLDMSK cloud mask (Frey et al. 2020) and CLDPROP cloud-top and optical property
(Platnick et al. 2021) continuity algorithms were developed. By applying common algorithms to a
subset of channels available on both instruments, the continuity algorithms accommodate the
detailed channel differences between the two instruments while maximizing the information
content on cloud parameters.
The continuity CLDPROP products have direct heritage with the Collection 6.1 MODIS
atmosphere cloud retrievals (MYD06), with cloud-top property datasets provided by the CLouds
from AVHRR (the Advanced Very High Resolution Radiometer) - Extended (CLAVR-x)
processing system (Heidinger et al. 2012, 2014). CLAVR-x produces cloud phase reported as
Cloud_Phase_Cloud_Top_Properties in the MODIS-VIIRS continuity cloud products. It replaces
the MODIS $CO_2$ slicing solution for cloud top pressure retrievals for cold clouds with an IR-
window channel optimal estimation approach coupled with a Cloud-Aerosol Lidar and Infrared
Pathfinder Satellite Observations (CALIPSO)-derived *a priori*. As a result, the CLDPROP optical
property cloud phase algorithm (reported as Cloud_Phase_Optical_Properties) removes the
dependence on the cloud top solution method in MYD06. Differences in the look-up tables (LUT)
of spectral liquid cloud reflectance result in changes of effective particle size (Re) (Platnick et al.
2020) that, along with cloud optical depth (COD), are used to derive cloud water path. Differences
with the Collection 6.1 MODIS cloud retrieval algorithms, as well as inter-sensor differences



between MODIS and VIIRS, have been reported in detail in recent studies such as Frey et al. (2020)
and Platnick et al. (2021), which are based on granule comparisons and long-term mean statistics.

2.2 Simultaneous Nadir Observations (SNOs) of collocated satellites

The pixel-scale comparisons will use SNOs between *Aqua*-AIRS and *SNPP*-CrIS. These SNOs

contain pixel pairs of observations from the two instruments when they observe the same location
at approximately the same scan angle and time. The AIRS-CrIS SNOs used herein were originally
developed by the JPL Sounder Science Investigator Processing System (SIPS) for inter-calibration
of two sounders (Manning and Aumann 2015). In order to ensure a close match between the
instruments, the following criteria are used to identify candidate SNOs:

•   FOV centers between *Aqua*-AIRS and *SNPP*-CrIS are within 8 km;

•   Observations are made within 10 minutes;

•   Both instruments observe within 3.3° of nadir, which corresponds with +/- 1 FOR

of AMSU for AIRS or ATMS for CrIS.


2.3 Pixel-scale collocations of imagers and sounders:

Utilizing the multi-sensor capability at the pixel scale requires accurate and computationally

efficient collocation of sounder and imager measurements. Various collocation methods exist
(Schreier et al. 2010, Nagle and Holz 2009, Yue et al. 2013). In this study, the method developed
by Wang et al. (2016) is applied by matching the instantaneous multi-sensor observations directly
based on line-of-sight (LOS) pointing vectors, defined as the vector from the satellite position to
the Earth surface pixel location. The details of this method and its accuracy are discussed at length
in Wang et al. (2016).



In this study, the same collocation method is applied to both *Aqua* and *SNPP* to match the finer
resolution imager pixels (MODIS and VIIRS) within a given sounder FOV (AIRS and CrIS). The
LOS vectors are calculated using the geolocation datasets for different sensors, which contain
latitude, longitude, satellite range, satellite azimuth and zenith angles. Collocation is performed
using the criterion that the angular difference between the LOS vectors for sounder and imager
should be less than half of the sounder FOV size angle. The CrIS FOV is treated as a 0.963° circle
which corresponds to ~41% of the peak response and collects ~98% of total radiation falling on
the detector (Wang et al. 2013). AIRS has a FOV half-power width of 1.1° (Fishbein et al. 2001).
However, 0.963° is used for both AIRS and CrIS in the collocation. After obtaining collocation
indices, the L2 cloud properties from both the imagers and sounders are populated accordingly.
The high spatial resolution information from MODIS and VIIRS is retained using higher statistical
moments and frequency distributions of cloud properties retrieved by imagers within collocated
sounder FOV. These statistical metrics include the mean, standard deviation, skewness and
kurtosis of MODIS and VIIRS cloud properties, the occurrence frequency of cloud types and cloud
phase reported by the cloud mask and cloud thermodynamic phase variables, and joint histograms
on the COD and CTP two-dimensional space following the convention of the International Satellite
Cloud Climatology Project (ISCCP, Rossow and Schiffer 1999). In addition to summarizing fine
imager spatial information over a coarser resolution sounder instrument, these statistical metrics
physically describe a variety of cloud processes at both regional and global scales for a range of
cloud types in different climate regimes, which are particularly relevant to sub-grid cloud
parameterization in numerical models (e.g. Zhu and Zuidema 2009, Kawai and Teixeira 2010 and
2012, Kahn et al. 2017). The ISCCP-type of joint histograms have been widely used to dissect the



uncertainty of the cloud radiative forcing (e.g. Pincus et al. 2012) and climate feedback (e.g.
Zelinka et al. 2012, Yue et al. 2016 and 2019) by cloud regimes (e.g. Oreopoulos et al. 2016).

By combining the SNOs and the sounder-imager collocated datasets, a multi-sensor multi-

satellite investigation is conducted to evaluate, at pixel scale, the self-consistency of cloud
properties, to benchmark data continuity from the US polar-orbiting operational environmental
satellites.

**3. Results**

Both *Aqua* and *SNPP* are in the 1:30 PM local equatorial crossing time sun-synchronous polar

orbits, but at different altitudes. This altitude difference gives a ~2.667 day repeating pattern for
AIRS and *SNPP*-CrIS observations at the same location. Accordingly, the number of SNOs
between these two IR sensors varies with time and a large fraction are located at the high latitudes.
In this study, seven focus days in January 2016 are selected for their large numbers of SNO pairs
and the full operation for all four instruments. Table 2 lists the focus days and gives the number of
observations obtained on each day. Figure 1 shows the latitudinal distribution of the focus day
SNOs (black bars, y-axis on the left, Table 2). A significant number of observations (>2,500) are
available at all latitudes, including the midlatitudes and tropics where SNOs are harder to obtain.

Fig. 2 shows the latitudinal variations of cloud frequency and zonal mean ECF and COD based

on the data from the seven focus days. To determine the detection of clouds in the sounder FOV,
two threshold values of ECF are used: 0.05 (solid lines) and 0.01 (dash lines). For MODIS and
VIIRS, frequency of Cloudy, Uncertain cases as reported by the cloud mask variable is shown for
MYD06 (black), MODIS continuity (red), and VIIRS continuity (blue) cloud products. Although
it is difficult to directly compare the mean cloud properties retrieved by imagers and sounders,





AIRS V7 produces similar general patterns of latitudinal variation of cloud frequency with the
imager products, which shows peaks of cloud occurrence in the tropics and midlatitude storm
tracks, and troughs in the subtropics. However, CLIMCAPS V2 cloud retrievals do not show these
variations, and its mean ECF values are much lower than AIRS V7 at all latitudes. A higher
percentage of cloud frequency in the low latitude regions is reported by AIRS V7 than by imagers,
consistent with previous findings showing higher sensitivity of hyperspectral IR sounders to
optically thin clouds (Kahn et al. 2014, Yue et al. 2016). An increase of COD with latitude at mid
to high latitude regions is detected by imagers, compared to a nearly flat or even decreasing mean
ECF retrieved by the sounders. These differences will be further assessed in the following
discussions.

3.1 Clouds retrieved by hyperspectral IR sounders

In Fig. 1, overlapped with the SNO count histograms are the occurrence frequency of

sounder FOVs (colored lines, y-axis on the right) for four composites that satisfy the following
four conditions, respectively: ECF > 0.01(general cloudy condition), ECF ≤ 0.01 (clear or very
thin clouds), ECF > 0.8 (overcast or very thick clouds), and cases with successful CTP retrievals
(QC for CTP is 0 or 1). These ECF values are selected based on the relationships between clouds
and the IR sounder spectral information, as well as the retrieval uncertainty. The fraction of the
highest quality atmospheric state retrievals below clouds, obtained from IR spectral information,
decreases with higher ECF (Fetzer et al. 2006). The combination of IR and MW radiances can
facilitate the retrieval of vertically resolved temperature and humidity profiles up to ECF of
0.7~0.8 (Yue et al. 2011, Yue and Lambrigtsen 2020, Yue et al. 2021). The ECF of 0.01 is often
used as the threshold of cloud detection by IR sounders (e.g. Kahn et al. 2014). Moreover, it has



been shown that AIRS V7 cloud retrievals present higher uncertainty on thin, broken clouds and
cloud edges when ECF < 0.01 (Yue and Lambrigtsen 2020).

For each composite, the occurrence frequency is calculated as the percentage of AIRS or

CrIS FOVs with successful cloud retrievals that satisfy the composite condition relative to the
total number of FOVs in each latitudinal bin. The QC flags for each cloud parameter are reported
in the L2 products and used to determine whether the algorithm reports a successful cloud
retrieval (when QC = 0 or 1). Different colors are used to indicate retrieval algorithms for the
two sounders. Since AIRS V7 and CLIMCAPS retrieve cloud properties up to two cloud layers
over each IR sounder FOV, an effective CTP is calculated as the weighted mean CTP by the
ECF reported at each cloud layer.

These results show large differences between the AIRS V7 clouds with those from CLIMCAPS.

AIRS V7 produces a much larger number of cloudy observations (solid pink line in Fig. 1) and a
higher yield for CTP retrievals (dash dotted line, Fig. 1), except in the Antarctic region. The
magnitude of this difference reaches up to 30% over the Southern Hemisphere and the tropics.
Furthermore, AIRS V7 produces much more overcast or very thick clouds (dash lines, Fig. 1) but
fewer clear or very thin cloudy cases (dotted lines, Fig. 1) than CLIMCAPS, which is consistent
with smaller mean ECF and lower cloud frequency in the tropics and midlatitude storm track
regions by CLIMCAPS V2 in Fig. 2. As discussed previously, this is related to the differences
between the two algorithms for AIRS in cloud clearing and cloud retrieval QC, as well as the use
of different *a priori*. These differences are further evaluated in the following sections using the
imager observations.

Despite the differences of sensors, satellites, and spectral resolutions, the three CLIMCAPS

Version 2 retrievals evaluated in this study present similar latitudinal distributions of the cloud





property distribution and cloud detection. As seen from Fig. 1, CLIMCAPS-*Aqua* (green dotted
line) reports a higher percentage of clear or very thin cloudy cases than those for *SNPP* (yellow
dotted line for CLIMCAPS-*SNPP* FSR and purple for CLIMCAPS-*SNPP* NSR), especially in the
midlatitude region. Among the three CLIMCAPS products, CLIMCAPS-*Aqua* (green solid line)
reports fewer cloudy cases than CLIMCAPS-*SNPP* (yellow and purple solid lines) in midlatitudes,
but more cloudy cases in the tropics. The finer spectral resolution for CLIMCAPS-*SNPP* FSR
retrievals produces a higher percentage of cloudy FOVs than the coarser spectral resolution
radiances used by the NSR retrieval.

Figure 3 further characterizes the four IR sounder cloud retrievals using the joint distributions

of observations among different algorithms. It is known that larger uncertainty of both sounder
and imager retrievals exists over snow and ice covered surfaces (Chan and Comiso, 2013, Yue and
Lambrigtsen 2020), so in this comparison the data points located in regions poleward of 60° are
excluded. Cases are only included if both data products in the comparison (indicated by x- and y-
axes of the plot) report valid retrievals. The three CLIMCAPS retrievals (x-axes) are compared
with AIRS V7 (y-axes) for both ECF and CTP. The generally good agreement among the
algorithms and sensors, especially for CTP, is encouraging, which shows the robustness of these
products and consistency of information for clouds in hyperspectral IR sounders. However,
CLIMCAPS reports a large number of cases with ECFs between 0 and 0.1, for which AIRS V7
reports ECFs ranging from 0 (clear sky) and 1 (completely cloudy). This issue is further illustrated
in Fig. 4. For cases where CLIMCAPS-*Aqua* V2 retrieved ECF is less than 0.1, AIRS V7 (the
magenta line) shows two peaks in the ECF occurrence frequency. The first peak is located at V7
ECF < 0.1, indicating the two algorithms agree with each other in cloud amount detection. The
larger second peak shows that more than 25% of cases with CLIMCAPS ECF < 0.1 have AIRS



V7 ECF values of 0.8~0.9. As a result, the correlation coefficient ($r$) between ECF retrievals from
AIRS V7 and CLIMCAPS V2 is only 0.27, which increases to 0.79 when neglecting ECF < 0.1
observations.

A tighter agreement between CLIMCAPS V2 and AIRS V7 is seen for CTP retrievals as shown

by points densely located along the identity line in Fig. 3. The correlation coefficients between
CLIMCAPS-Aqua and AIRS V7 CTP are 0.69 for all cases and 0.92 for ECF > 0.1, respectively.
High cloud cases (AIRS V7 CTP < 440hPa) show a much higher CTP correlation (r = 0.87) than
for low clouds (AIRS V7 CTP > 600 hPa, $r$ = 0.43). When both algorithms identify low clouds in
the FOV, CLIMCAPS reports a slightly lower cloud top (larger CTP) than AIRS V7, with a median
value difference of 12 hPa; whereas for high clouds, CLIMCAPS V2 reports a higher cloud top
with its median CTP 13 hPa smaller than the one by AIRS V7.

In the next section, these differences among the various sounder cloud retrieval products are

further evaluated using the cloud parameters determined by collocated MODIS and VIIRS data.

3.2 Comparison of sounder cloud properties and collocated imager measurements

Figures 5 and 6 compare the cloud properties retrieved from various sounder algorithms with

the collocated imager cloud retrievals in the MYD06 and CLDPROP_MODIS products,
respectively. Comparisons with CLDPROP_VIIRS are similar to those using CLDPROP_MODIS
and hence are not shown in these figures. The cloud properties from MODIS pixels are averaged
within the collocated sounder FOV before this comparison.

The IR sounder retrieved ECF is positively correlated with the imager observed COD in the

top rows of Figs. 5 and 6, showing the consistency of cloud amount determined using different
sensors. However, two main differences are noticed. First, it is clear that the CLIMCAPS V2 (for





both *Aqua* and *SNPP*) misidentifies a significant number of cloudy cases as clear or thin clouds.
As shown in Fig. 4, more than 50% of these cases are optically thick clouds with large cloud
amount (ECF > 0.7) reported by AIRS V7 and COD values ranging from 2 to 10 by MODIS and
VIIRS. Secondly, the comparisons between CLIMCAPS and imager cloud products do not have
the cluster corresponding to cases with both high ECF and large COD values, as in the comparison
between AIRS V7 and imagers. As discussed previously, this is related to misidentification of
cloudy cases as clear or thin cloud conditions by CLIMCAPS. However, another main cause is
that CLIMCAPS cloud retrievals have the same QC flags as the physical atmospheric state
retrievals; as a result, cases with large cloud amount are filtered out. In general, AIRS V7 products
exhibit better agreement with MODIS and VIIRS in detecting cloud amount and occurrence.
CLIMCAPS V2 cloud retrievals could be further improved with better cloud clearing flow and
more careful treatment when retrieving clouds with unsuccessful atmosphere physical retrievals.

The sounder and imager CTP retrievals are also compared in the bottom rows of Fig. 5 and 6.

Despite instrument and algorithm differences, when both sounder and imager detect high clouds
(CTP < 440 hPa, including ECF < 0.1 cases), CTP retrievals agree with each other well. The
correlation coefficients with MYD06 CTP are 0.77, 0.52, and 0.62 for AIRS V7, CLIMCAPS-
*Aqua*, and CLIMCAPS-*SNPP*-FSR, respectively. When imagers detect low clouds (CTP > 680
hPa), IR sounders determine the majority of cases as low clouds but with a tail toward CTP values
corresponding to high and mid-level clouds (middle row). The disagreement mainly occurs when
sounder retrieved ECF is less than 0.1 as shown by the magenta contour lines. These are cases
when larger uncertainty in infrared cloud retrieval exists, as discussed previously. After removing
these cases, the sounder-imager discrepancy in the low cloud conditions is reduced greatly (bottom
row), especially for AIRS V7. These differences are consistent with the known limitation of



imagers such as MODIS, which tend to miss high and thin cloud layers (Holz et al. 2008) when
compared with AIRS (Kahn et al. 2014). However, the analysis presented here cannot completely
rule out the impact of uncertainty in the IR sounder cloud retrievals. When both hyperspectral
sounders and narrowband imagers detect low clouds, sounders tend to retrieve smaller CTP than
imager. For AIRS V7, the median difference in this condition is -65, -77, and -80 hPa with MYD06,
CLDPROP_MODIS, and CLDPROP_VIIRS products, respectively.

3.3 Clouds retrieved by imagers

Figure 7 compares COD, CTP, and Re retrieved by different MODIS and VIIRS cloud

algorithms, with mean imager cloud properties over corresponding sounder FOVs are shown. Very
good agreement between MODIS and VIIRS, and between the MYD06 and continuity products is
seen. All correlation coefficients are greater than 0.8. For the three cloud parameters, correlation
is always the highest between products derived from the same instrument (MYD06 and
CLDPROP_MODIS), and the lowest between MYD06 and CLDPROP_VIIRS (but still reaching
0.81, 0.88, and 0.81 for COD, CTP, and Re, respectively) when both instrument and algorithm are
different. From the same instrument MODIS but different algorithms, the correlation is lowest for
CTP retrievals (r = 0.89) compared to COD (r = 0.97) and Re (r = 0.97). This is because MYD06
and the continuity cloud algorithm uses different methods and spectral channels to determine CTP.
However, a relationship near one-to-one is seen, indicating the consistency between the
operational and continuity cloud products from MODIS, at least for the cloud properties averaged
at the sounder resolution (~13.5km). Correlations between MODIS and VIIRS cloud products are
lower than those from MODIS alone (with different algorithms), even when both products are
derived from the same continuity algorithm. The degradation of agreement is larger for COD and



Re than for CTP (Fig. 6). This reflects the effect of spectral channel and spatial resolution
differences between MODIS and VIIRS, as well as the related adjustments made to the continuity
algorithms, such as the liquid phase LUT for cloud microphysical retrievals. Another possible
factor is the collocation error existing in the SNOs, but this is ruled out since results with more
conservative collocation criteria remain largely the same (not shown).

To further analyze the differences between the imager cloud products and the subpixel cloud

heterogeneity over the sounder FOVs, the standard deviation and skewness of the imager cloud
property distributions over the sounder FOVs are shown in Fig. 8 and 9, respectively. Correlations
are weaker in these higher statistical moments, yet for standard deviation they remain larger than
0.6. Similar to comparisons for mean values, tight one-to-one relationships are seen for standard
deviation at the sounder FOV scale between the two MODIS cloud products. Similar to mean value
comparisons, the CTP standard deviation has the lowest correlation coefficient ($r = 0.63$) compared
to the ones for COD ($r = 0.96$) and Re ($r = 0.87$).  However, skewness only shows significant
correlations for COD ($r = 0.78$) and Re ($r = 0.70$) between the two MODIS datasets, but poor
correlations ($r < 0.3$) for CTP. The impact from the differences in CTP algorithms thus shows up
more strongly on the higher statistical moments. When evaluating data from different sensors, no
correlation is seen for skewness of any of the cloud parameters even with the same retrieval
algorithms (Fig. 9, middle and right columns), different from the comparisons using mean value
and standard deviation (Figs. 7 and 8, middle and right columns).

3.4 Joint histograms, cloud types, and cloud thermodynamic phase
3.4.1   Cloud type by cloud property joint histograms



Figs. 10-13 show the two-dimensional cloud histograms calculated using SNOs from the focus
days over different surface types and regions, including the tropics (30°N~30°S), over ocean (land
fraction < 0.1, 60°N~60°S), over land (land fraction > 0.9, 60°N~60°S), and over ice and snow
covered surfaces (frozen surfaces), respectively. The land fraction and surface classes are obtained
from the AIRS V7 L2 product under variable names of landFrac and SurfClass, respectively. For
MODIS and VIIRS, the ISCCP type of CTP-COD joint histograms are generated by summing the
joint distributions over individual AIRS and CrIS FOV, with no averaging over sounder FOV. For
AIRS and CrIS, joint distributions are calculated on the CTP and ECF space.
Consistent with results in previous sections, AIRS V7 shows peaks of both thin and thick
clouds while CLIMCAPS V2 products show a single peak distribution of thin clouds. Better
consistency of AIRS V7 with imager cloud products is also shown by the joint histograms. For
example, in the tropics (Fig. 10) clusters corresponding to optically thick high clouds, thin cirrus,
and broken or optically thin low clouds are seen in the AIRS V7 CTP-ECF histogram, consistent
with the patterns in the MODIS and VIIRS CTP-COD histograms. Agreement between AIRS V7
and imager clouds is also found for mid-level and low cloud clusters over ocean (Fig. 11) and for
high and mid-level clouds over land (Fig. 12). Over frozen surfaces (Fig. 13), the sounder clouds
show optically thin and high clouds, especially in CLIMCAPS V2; a large percentage of mid-level
clouds with medium to large ECF values are seen in AIRS V7, more consistent with the cloud
histograms from imager observations. However, MODIS and VIIRS cloud detection and retrievals
suffer a higher uncertainty over frozen surfaces (Chan and Comiso, 2013), and the small
atmospheric thermal contrast with frozen surfaces presents additional challenges for hyperspectral
IR sounder retrievals (Yue and Lambrigtsen 2020). Therefore, more accurate cloud measurements



from in-situ or active space-borne instruments are needed to further quantify the quality of these
imager and sounder cloud retrieval products in snow- and ice-covered regions.

Because of its long temporal coverage since 1999 when *Terra* MODIS began operating, high

quality, and the distinct physical characteristics of different cloud types, the MODIS cloud data
record, especially the CTP-COD joint histograms, have been widely used in different aspects of
climate studies. These include detailed analyses on the radiative effect of different cloud types
(Yue et al. 2016, Oreopoulos et al. 2016), evaluation of climate model simulations of clouds
(Pincus et al. 2012), quantification of the cloud feedback by different cloud types (Zhou et al. 2014,
Yue et al. 2019), and investigations of cloud impacts on hydrological cycle and the global
circulation (Su et al. 2017), especially in the tropics. Therefore, the differences of the cloud
frequency histograms from various imager retrieval products in the tropics are further analyzed
here. In Fig. 14, the MODIS continuity product (depicted in Fig. 10) is used as the common base
to evaluate the differences caused by algorithms and sensors: 1) between current NASA standard
MODIS retrievals and the MODIS continuity algorithms, and 2) between the MODIS and VIIRS
continuity cloud data records. The magnitude of joint frequency histogram differences is within
±5% using the focus day observations. MYD06 shows more clouds with CTP < 180 hPa but fewer
low clouds with CTP > 800 hPa than the continuity product, consistent with findings in Platnick
et al. (2021). VIIRS continuity cloud retrievals produce higher frequencies of clouds with COD
between 9.4 and 60, but fewer high clouds with COD < 9.4. Whether and how these differences
will impact the long-term trend and short-term variability of clouds as seen by the imagers warrants
further study.
3.4.2   Cloud thermodynamic phase



Both MYD06 and continuity cloud products provide cloud thermodynamic phases (Table 1),
given by the optical property retrieval (Cloud_Phase_Optical_Properties, in both MYD06 and
continuity products) and the CLAVR-x processing system (Cloud_Phase_Cloud_Top_Properties,
continuity products only). The Cloud_Phase_Cloud_Top_Properties variable reports flags
determining pixels to be cloud free, water cloud, ice cloud, mixed phase cloud, or undetermined
phase. The Cloud_Phase_Optical_Propertes flags indicate cloud mask not determined for pixel,
clear sky, liquid water cloud, ice cloud, or undetermined phase, the last of which includes mixed
phase clouds (Marchant et al. 2016). AIRS thermodynamic cloud phase, which is available in the
AIRS V6 and V7 Level 2 Support product, is based on a set of brightness temperature difference
and threshold tests using the channels in 960, 1231, 930, and 1227 cm$^{-1}$ (Nasiri and Kahn 2008,
Kahn et al. 2014). These tests are applied to AIRS FOVs where ECF > 0.01, and classify the AIRS
FOV as containing liquid, ice, or unknown cloud phases. Detailed comparisons of AIRS cloud
phases with CALIPSO indicate good agreement with CALIPSO on ice phase detection, and
conservative liquid phase determination (Jin and Nasiri 2014, Peterson et al. 2020). These studies
also show that the unknown class of AIRS cloud phase corresponds to scenes containing both ice
and liquid particles, and low-level liquid clouds, especially in the trade-wind cumulus cloud regime.
Figs. 15-18 show the histograms of cloud thermodynamic phase (solid color bars for imagers
and magenta symbols for AIRS) for the same set of focus-day SNOs. Similar to joint histograms
in Fig. 10-13, each figure shows results over the four types of surfaces and regions: tropics, ocean,
land, and frozen surfaces. MODIS and VIIRS cloud mask histograms (hollow color bars) are also
shown in the figures, together with the frequency of clear sky detected by IR sounders (ECF <
0.01, colored solid circles). Note that for MODIS and VIIRS, the mixed-phase or undetermined
phase category is shown with the y-axis on the right due to their much smaller frequency of



occurrence. For clear sky detection, the cloud-mask clear frequencies from all the imager products
are similar except over the frozen surfaces, where VIIRS cloud mask shows 10% higher frequency
than MODIS. For IR sounders, AIRS V7 produces significantly lower clear-sky frequency than
CLIMCAPS and imager cloud products over non-frozen surfaces. Over frozen surfaces, more
frequent clear conditions are reported by AIRS V7 than CLIMCAPS, although AIRS V7 is more
consistent with the clear frequency from MODIS and VIIRS data.

The frequencies of liquid or ice phase clouds are highly consistent between two cloud phase

variables in various imager cloud products, except for ice phase determination over frozen surfaces.
This is supported by the low uncertainty range of ice and liquid phase for these four conditions as
shown in Table 3. Here uncertainty is roughly characterized by the standard deviation of estimates
from different products and variables. The Cloud_Phase_Cloud_Top_Properties reports higher
percentage of liquid phase than Cloud_Phase_Optical_Propertes. In particular, the VIIRS cloud
top cloud phase product always reports the highest frequency of liquid clouds. From both cloud
phase variables, MODIS reports more ice and fewer liquid clouds than VIIRS. When looking at
Cloud_Phase_Optical_Propertes for MODIS, ice (liquid) cloud frequency is higher (lower) in
MYD06 than in the CLDPROP_MODIS products. The undetermined phase by the
Cloud_Phase_Optical_Propertes includes both mixed and uncertain phases (Baum et al. 2012).
Except in tropics, MYD06 has the higher frequency of undetermined cases than the continuity
cloud products, and this is most prominent over the frozen surfaces with MYD06 reporting ~2.8%.

AIRS cloud phase retrievals report a higher frequency of ice clouds than imagers under all

conditions, especially in the tropics (Fig. 15) and over land (Fig. 17). However, a much lower
frequency of liquid clouds is retrieved by AIRS, which is consistent with a more conservative
liquid phase determination approach applied by AIRS cloud phase algorithm (Kahn et al. 2014).



The unknown phase of AIRS ranges from ~15% over the frozen surfaces to ~45% over ocean and
in the tropics, which corresponds with broken and thin low clouds and scenes with both ice and
liquid cloud particles (Jin and Nasiri 2014).

**4.   Summary**

In this study, the pixel-scale collocation between the hyperspectral infrared (IR) sounders

(AIRS and CrIS) and high spatial resolution imagers (MODIS and VIIRS) is performed on the
pairs of Simultaneous Nadir Observations (SNOs) between *Aqua*-AIRS and *SNPP*-CrIS. Using
this approach, the cloud parameters retrieved by various algorithms for IR sounders and imagers
from different platforms are evaluated at the pixel level. Quantifying uncertainty in the cloud
observational data records is important for constraining the high uncertainty of clouds in weather
and climate research. This is also crucial in improving the retrieval of atmospheric, surface, and
radiation properties since satellite observations are highly subject to uncertainties and limitations
associated with cloud conditions in the instrument field of view (FOV) (e.g. Yue et al. 2013, Wong
et al. 2015, Tian et al, 2020). Moreover, narrowband imagers and hyperspectral sounders provide
important components of the long-term sustained observations of cloud properties in the Program
of Record (POR), as noted by the 2017 US National Academy Decadal Survey (ESAS 2017). The
analyses presented here will help to assess the capability of the POR, thus to identify potential
gaps existed in the POR for cloud properties.

Both the NASA standard and continuity retrieval algorithms for sounders and imagers are

investigated here in order to quantify the differences among the retrieval products, and to examine
the consistency and continuity of the data products from multiple sensors across different satellites.
This is essential to the goal of building a continuous record of satellite data using the *Terra*, *Aqua*,





*SNPP*, and *JPSS* series satellites, with sufficient quality to detect and quantify global
environmental change.
Multiple cloud parameters are analyzed (Table 1). Comparisons are made by investigating the
mean cloud parameters, and higher statistical moments of cloud property distributions measured
by MODIS and VIIRS over the corresponding AIRS and CrIS FOV. Cloud types indicated by the
joint histograms of cloud properties and cloud thermodynamic phases are included. Through these
comparisons, good agreement is found between the sounder and imager retrieved cloud products,
yet with distinct differences likely arising from algorithm and sensor differences. For IR sounders,
cloud top pressure (CTP) retrieved by AIRS Version 7 (V7) and CLIMCAPS (-*Aqua* and -*SNPP*)
Version 2 (V2) agree, as shown by correlation coefficients of 0.69 for all cases and 0.92 for cases
with effective cloud fraction (ECF) greater than 0.1, respectively. Compared to AIRS V7,
CLIMCAPS tends to produce a lower cloud top (CTP 12 hPa larger) for low clouds, but higher
cloud top (CTP 13 hPa smaller) for high clouds. However, CLIMCAPS V2 significantly
overestimates the frequency of clear and optically thin cloud (ECF < 0.1), relative to AIRS V7 and
imager products from both MODIS and VIIRS. This is due to the algorithmic differences between
CLIMCAPS V2 and AIRS V7 cloud retrieval algorithms. These differences include whether
iteration of cloud clearing is performed, the surface/atmospheric states used in the cloud retrieval,
the quality control procedures used, and different *a-priori* states used by AIRS V7 and CLIMCAPS.
How these differences affect the downstream atmospheric and surface retrievals in the two
algorithms, and the attribution of impacts from each factor, is beyond the scope of this study and
warrants further investigation.
High consistency is seen among different imager cloud products, especially in the mean and
standard deviation of cloud properties from the MODIS atmosphere cloud property retrieval





(MYD06) and the MODIS-VIIRS continuity cloud products (CLDPROP). The magnitude of the
correlation coefficients closely reflects the impact of algorithm differences and instrument spectral
and resolution differences, with highest correlations obtained between two MODIS products (same
sensor but different algorithms) and lowest between MYD06 and CLDPROP_VIIRS (different
sensors, different algorithms). The correlation coefficients are always higher for cloud optical
depth (COD) and particle effective radius (Re) than for CTP. For mean cloud properties, they are
as large as 0.97 between MYD06 and CLDPROP_MODIS, and 0.89 for CTP. For standard
deviations within the sounder FOV, the correlations are smaller than those for mean cloud
properties, ranging from 0.77 to 0.96 for COD, 0.66 to 0.97 for Re, but only 0.60 to 0.63 for CTP.
This is likely due to the fact that completely different CTP retrieval methods are used in the
MODIS operational and continuity cloud algorithms to accommodate the lack of near-IR and IR
water vapor and $CO_2$ absorption channels in VIIRS. Such algorithm and instrument impacts are
more apparent in the higher moment statistics of cloud properties such as skewness. The
correlations of COD and Re skewness between MYD06 and CLDPROP_MODIS drop to 0.78 and
0.70, respectively. They are further reduced to below 0.4 when comparing MODIS and VIIRS
cloud products. For CTP skewness, the correlation coefficients are less than 0.3.

Two different cloud thermodynamic phase retrievals are available from imager observations,

which are obtained by the optical property retrieval (Cloud_Phase_Optical_Properties, in both
MYD06 and MODIS-VIIRS continuity products) and the CLAVR-x processing system
(Cloud_Phase_Cloud_Top_Properties, continuity products only). The frequencies of liquid or ice
phase clouds are very consistent between two cloud phase variables in different imager cloud
products, with uncertainty usually generally less than 4%. The largest uncertainty is reported for
ice phase determination over snow and ice covered surfaces. MODIS retrievals report more ice

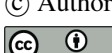



and fewer liquid clouds than VIIRS, consistent with findings by Platnick et al. (2020). Comparing
the two different cloud phase retrievals, the Cloud_Phase_Cloud_Top_Properties reports higher
percentages of liquid phase than Cloud_Phase_Optical_Properties, and the
Cloud_Phase_Optical_Properties in MYD06 detects higher (lower) frequencies of ice (liquid)
clouds than that in the CLDPROP_MODIS products.
The general consistency of cloud observations among different sensors aboard *Aqua* and *SNPP*
from various algorithms is encouraging, especially for achieving a continuous multi-decadal
climate data record of clouds that can extend beyond the A-Train era and well into the 2030s with
the *JPSS* series. The quantification of algorithm differences has important implications for future
retrieval algorithm developments, and will further improve the capability and accuracy of such
climate data records.

**Data and Code Availability:**
MODIS (MYD06 10.5067/MODIS/MYD06_L2.061; MYD35
10.5067/MODIS/MYD35_L2.061; CLDPROP-MODIS
10.5067/VIIRS/CLDPROP_L2_MODIS_Aqua.011; CLDMSK-MODIS
10.5067/MODIS/CLDMSK_L2_MODIS_Aqua.001) and VIIRS data (CLDPROP-VIIRS
10.5067/VIIRS/CLDPROP_L2_VIIRS_SNPP.011; CLDMSK-VIIRS
10.5067/VIIRS/CLDMSK_L2_VIIRS_SNPP.001) were obtained through the Level-1
Atmosphere Archive and Distribution System (LAADS; http://ladsweb.nascom.nasa.gov/). AIRS
(AIRS V7 Level 2 Support Product 10.5067/APJ6EEN0PD0Z; CLIMCAPS-Aqua Version 2
Level 2 10.5067/JZMYK5SMYM86) and CrIS data (CLIMCAPS-SNPP Version 2 FSR
10.5067/62SPJFQW5Q9B; CLIMCAPS-SNPP Version 2 NSR 10.5067/8RUZI1F8U1UX) were



obtained from the NASA Goddard Earth Sciences Data Information and Services Center
(GESDISC) and could be accessed at https://earthdata.nasa.gov/. The collocation code is publicly
available from https://github.com/wanglikun1973/CrIS_VIIRS_collocation. The data used to
generate the figures and tables in this study can be obtained by contacting the corresponding
author.

**Author Contribution:**
QY conceptualized the study, developed the methodology and datasets, carried out the formal
analyses, and contributed to the writing of the manuscript. EF, BK, NS, JB, and BL contributed
to the data curation, validation, investigation, and the writing of the manuscript. LW, IT, MM,
and KM contributed to the data curation and software.

**Competing Interests:**
The authors declare that they have no conflict of interest

**Acknowledgements:**
The research was carried out at the Jet Propulsion Laboratory, California Institute of
Technology, under a contract with the National Aeronautics and Space Administration
(80NM0018D0004). QY, EJF, BHK, and BL were supported by NASA's Making Earth Science
Data Records for Use in Research Environments (MEaSUREs) program. QY was supported by
the NASA CloudSat and CALIPSO Science Team Recompete NNH15ZDA001N-CCST grant.
QY, EJF, MS, and BHK acknowledge the support of the AIRS Project at JPL and the sounder
SIPS at JPL.





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





Table 1: The satellite cloud parameters examined in this study, and the retrieval algorithms
and products from which these parameters are obtained.

| Satellite | Sensor | Retrieval Algorithm / Product | Cloud Parameters |
|---|---|---|---|
| *Aqua* | AIRS | AIRS Version 7 Level 2 Standard and Support Product | • Effective Cloud Fraction (ECF)<br>• Cloud Top Pressure (CTP)<br>• Cloud Thermodynamic Phase |
| | | Version 2 CLIMCAPS-*Aqua* Level 2 Infrared and Microwave Combined Retrieval | • Effective Cloud Fraction (ECF)<br>• Cloud Top Pressure (CTP) |
| | MODIS | Collection 6.1 *Aqua* MODIS Atmosphere Level 2 Cloud Product (MYD35, MYD06) | • Cloud Mask<br>• Cloud Top Pressure (CTP)<br>• Cloud Optical Depth (COD)<br>• Cloud Effective Radius (Re)<br>• Cloud Phase Optical Properties |
| | | Version 1.1 NASA MODIS Continuity Cloud Mask and Cloud Property Products (CLDMSK/CLDPROP_MODIS) | • Cloud Mask<br>• Cloud Top Pressure (CTP)<br>• Cloud Optical Depth (COD)<br>• Cloud Effective Radius (Re)<br>• Cloud Phase Optical Properties<br>• Cloud Phase Cloud Top Properties |
| *SNPP* | CrIS | Version 2 CLIMCAPS-*SNPP* FSR Level 2 Retrieval | • Effective Cloud Fraction (ECF)<br>• Cloud Top Pressure (CTP) |
| | | Version 2 CLIMCAPS-*SNPP* NSR Level 2 Retrieval | • Effective Cloud Fraction (ECF)<br>• Cloud Top Pressure (CTP) |
| | VIIRS | Version 1.1 NASA VIIRS Continuity Cloud Mask and Cloud Property Products (CLDMSK/CLDPROP_VIIRS) | • Cloud Mask<br>• Cloud Top Pressure (CTP)<br>• Cloud Optical Depth (COD)<br>• Cloud Effective Radius (Re)<br>• Cloud Phase Optical Properties<br>• Cloud Phase Cloud Top Properties |






Table 2 Number of SNOs between *Aqua*-AIRS and *SNPP*-CrIS on the seven focus days used
in this study.

| Focus Day | Jan. 01, 2016 | Jan. 03, 2016 | Jan 04, 2016 | Jan 09, 2016 | Jan 11, 2016 | Jan 14, 2016 | Jan 17, 2016 |
|---|---|---|---|---|---|---|---|
| # of SNOs | 10,000 | 10,000 | 1372 | 10,000 | 10,000 | 10,000 | 8,903 |







Table 3. The mean value and uncertainty range of the occurrence frequencies of ice and liquid
phase clouds based on the cloud thermodynamic phase variables from the three imager cloud
retrievals: MYD06, CLDRPOP_MODIS, and CLDPROP_VIIRS. Results over the five types of
surfaces and regions are shown respectively for tropics, ocean, land, frozen surfaces, and global.
For each condition, five estimates of cloud phase frequencies are available based on two types of
imager-derived cloud thermodynamic phase: Cloud_Phase_Optical_Properties determined by the
optical property retrieval (provided in both MYD06 and the two continuity products), and
Cloud_Phase_Cloud_Top_Properties obtained through the CLAVR-x processing system applied
in the continuity cloud algorithm (provided in the CLDPROP-MODIS and -VIIRS cloud
products). The uncertainty range is characterized by the standard deviation of the five estimates
obtained in each region.

| Frequency (%) | Tropics (30°N~30°S) | 60°N~60°S Ocean | 60°N~60°S Land | Frozen Surfaces | Global, All Cases |
|---|---|---|---|---|---|
| Liquid Phase | 37.64±3.21 | 53.94±3.50 | 35.16±2.81 | 14.03±1.10 | 44.27±2.79 |
| Ice Phase | 26.36±1.96 | 21.32±2.59 | 23.37±1.03 | 14.28±4.38 | 20.43±3.02 |


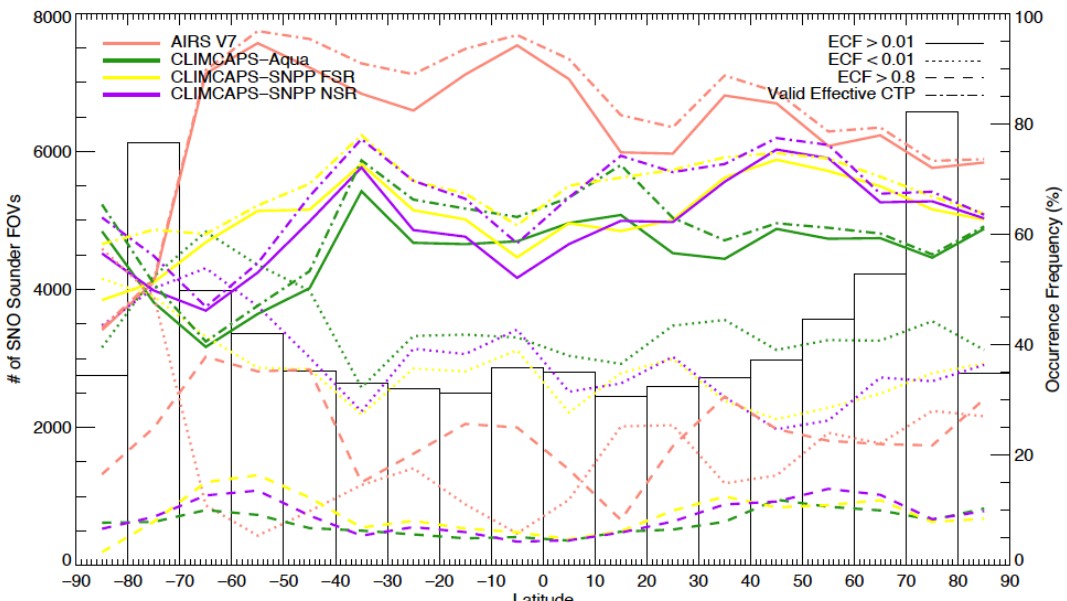

Figure 1. The latitudinal distribution of the SNO pairs for *Aqua*-AIRS and *SNPP*-CrIS (black
bars) and the occurrence frequencies of various sounder retrieved cloud parameters (right y-
axis, %) for four composites that satisfy the following four conditions, respectively: ECF >
0.01(solid lines, general cloudy condition), ECF ≤ 0.01 (dotted lines, clear or very thin clouds),
ECF > 0.8 (dash lines, overcast or very thick clouds), and cases with successful CTP retrievals
(dash dotted lines, QC for CTP is 0 or 1). Data from the seven focus days are used (see Table 2)
and binned by latitude of the sounder FOVs in 10° latitude bins. Four different sounder retrieval
products are shown by colored lines: AIRS Version 7 (AIRS V7, pink), CLIMCAPS-*Aqua*
(green), CLIMCAPS-*SNPP* FSR (yellow), and CLIMCAPS-*SNPP* NSR (purple). Occurrence
frequency is calculated as the percentage of AIRS or CrIS FOVs with successful cloud retrievals
(quality control indicator = 0 or 1) satisfying the aforementioned four conditions to the total
number of FOVs in each latitudinal bin.



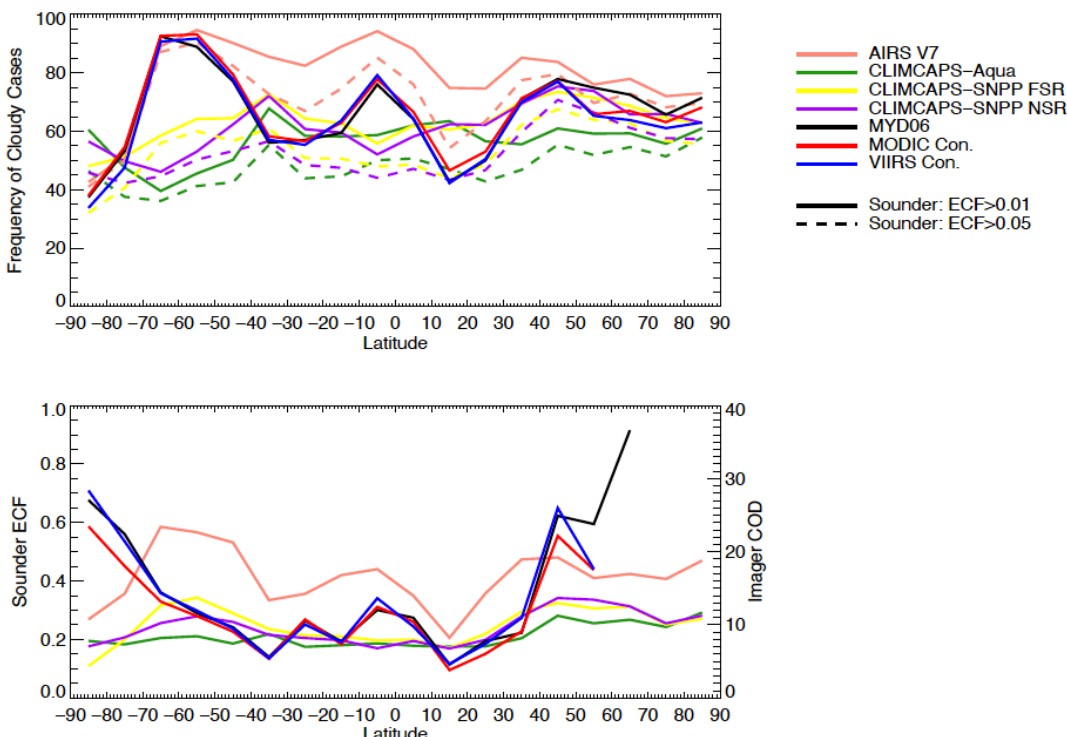

Figure 2. a) Zonal mean frequency of cloudy cases as observed by hyperspectral sounders and imagers. For MODIS and VIIRS, frequency of Cloudy, Uncertain cases as reported by cloud mask is shown for MYD06 (black), MODIS continuity (red), and VIIRS continuity (blue) cloud products. For AIRS and CrIS, solid and dash lines show frequencies of sounder FOVs with ECF > 0.01 and ECF > 0.05, respectively. Results for AIRS Version 7 (AIRS V7, pink), CLIMCAPS-*Aqua* (green), CLIMCAPS-*SNPP* FSR (yellow), and CLIMCAPS-*SNPP* NSR (purple) are shown for sounder cloud products. b) Zonal mean values of sounder ECFs (left y axis) and imager COD (right y axis) from these retrieval algorithms.



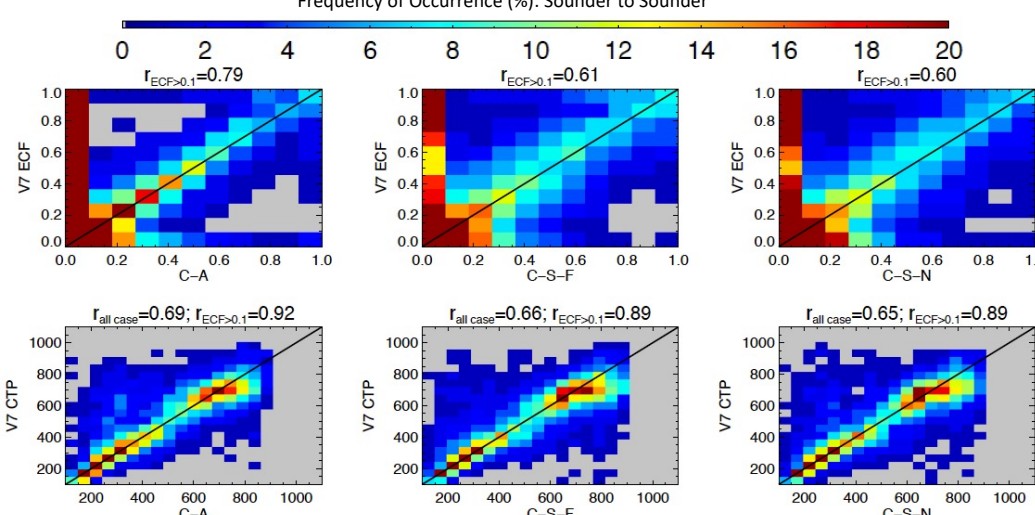

Figure 3. Comparisons of ECF (top row) and effective CTP (bottom row) derived from different
sounder retrieval algorithms. Linear correlation coefficients are calculated for cloud properties
obtained from retrieval products indicated on the axes and are given on top of the each plot.
From left to right, results comparing AIRS Version 7 with CLIMCAPS-*Aqua* (C-A),
CLIMCAPS-*SNPP* FSR (C-S-F), and CLIMCAPS-*SNPP* NSR (C-S-N) are shown using joint
distributions of frequency of occurrence (%). The data points located in regions poleward of 60°
are excluded. Cases are included only when both retrievals in comparison (x- and y-axes of the
plot) report valid retrievals.





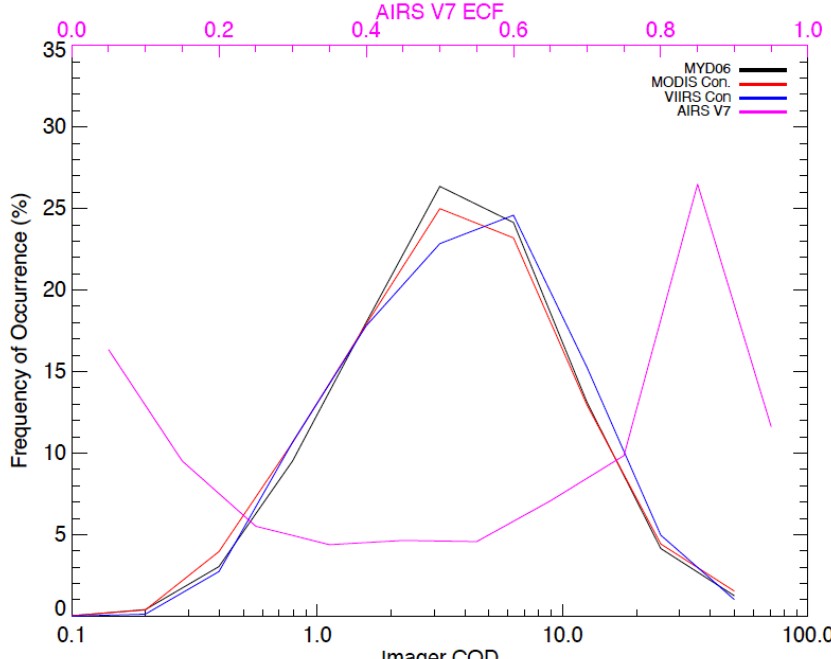

Figure 4. Frequency histograms showing the density distributions of imager cloud optical depth (COD, bottom x-axis) and AIRS V7 ECF (magenta, upper x-axis) for cases where V2 CLIMCAPS-*Aqua* retrieves an ECF value less than 0.1. Different imager cloud products are included: MYD06 (black), *Aqua*-MODIS continuity cloud products (MODIS Con., red), and *SNPP*-VIIRS continuity cloud products (VIIRS Con., blue).

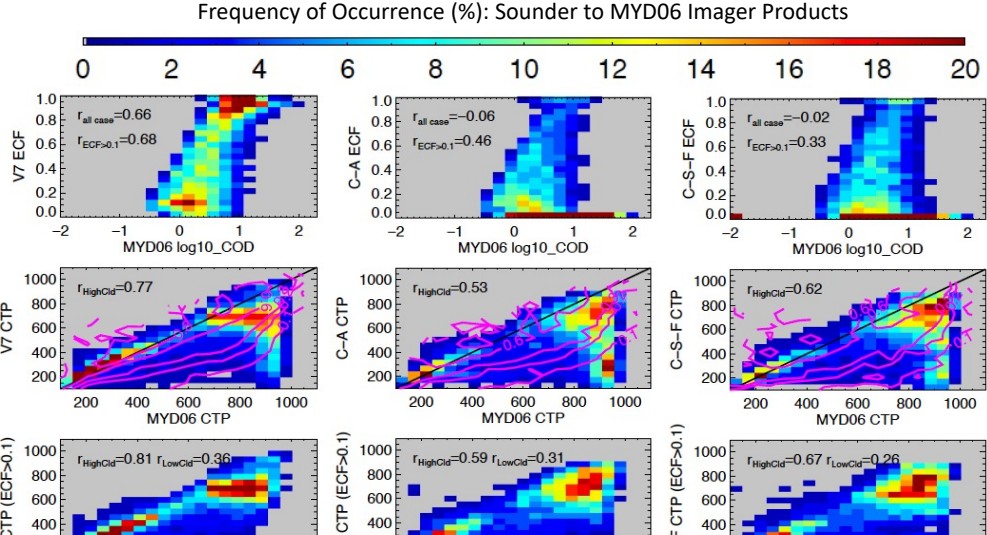

Figure 5. Comparisons of sounder and imager derived cloud properties shown by joint
distribution of case frequency of occurrence. Top row shows evaluation of sounder-derived ECF
by cloud optical depth (COD, in log10 scale) from the MYD06 products. The middle row
compares the sounder effective CTP with CTP from MYD06 overlaid by the magenta contours
showing the mean ECF from the corresponding sounder retrievals. The bottom row is similar to
the middle row except that the cases with sounder ECF < 0.1 are removed from the comparison.
Different sounder retrieval algorithms are included. From left to right, data from AIRS Version
7, CLIMCAPS-*Aqua* (C-A), and CLIMCAPS-*SNPP* FSR (C-S-F) are used. The data points
located in regions poleward of 60° are excluded. Cases are included only when both retrievals in
comparison (x- and y-axes of the plot) report valid retrievals. The cloud properties from MODIS
pixels collocated within the same sounder FOV are averaged before comparing with the IR
sounder data. Linear correlation coefficients between the variables on x- and y-axes for different
conditions are given in each plot.





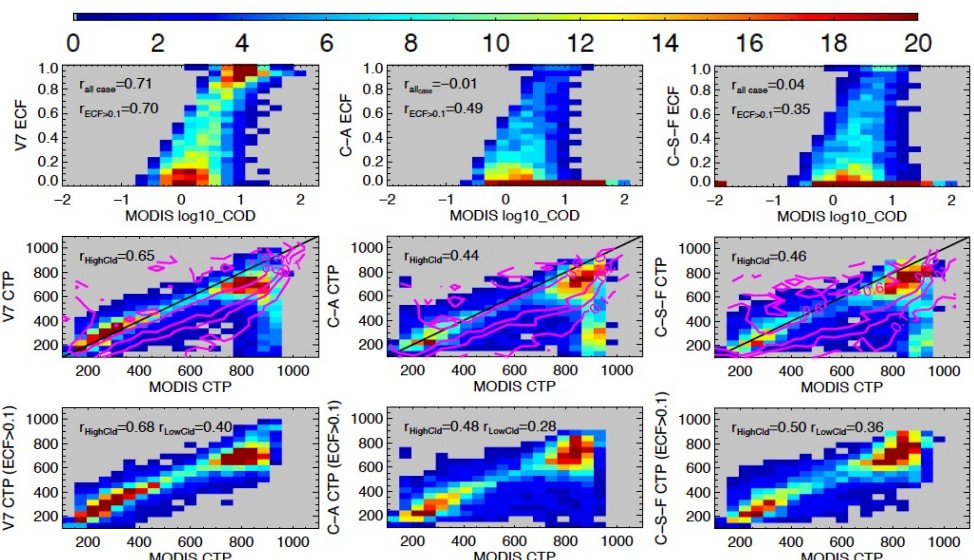

Figure 6. Similar to Fig. 5, except using the MODIS continuity cloud product
(CLDPROP_MODIS).



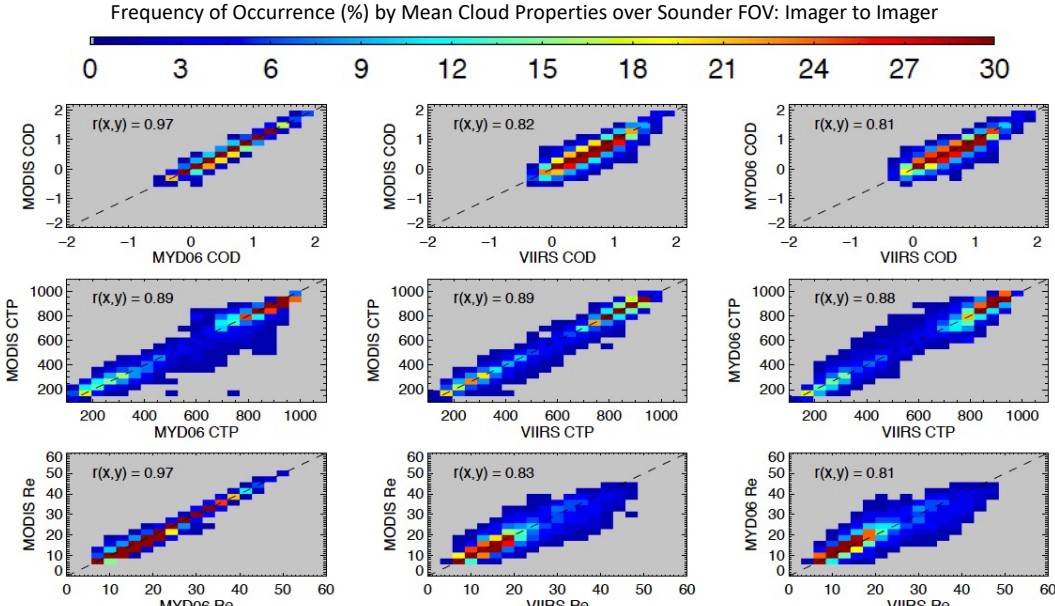

Figure 7. Comparison of cloud optical depth (COD, in log10 scale), cloud top pressure (CTP, hPa), and effective particle size (Re, $\mu$m) retrieved by MODIS and VIIRS cloud algorithms. The mean imager cloud properties over corresponding sounder FOVs are compared over the SNOs. From left to right show the results of following comparisons: *Aqua* MODIS continuity cloud products (CLDPROP_MODIS) with MYD06, CLDPROP_MODIS with *SNPP*-VIIRS continuity cloud products (CLDPROP_VIIRS), and MYD06 with CLDPROP_VIIRS, respectively. Linear correlation coefficients between the variables on x- and y-axes are given in each plot.

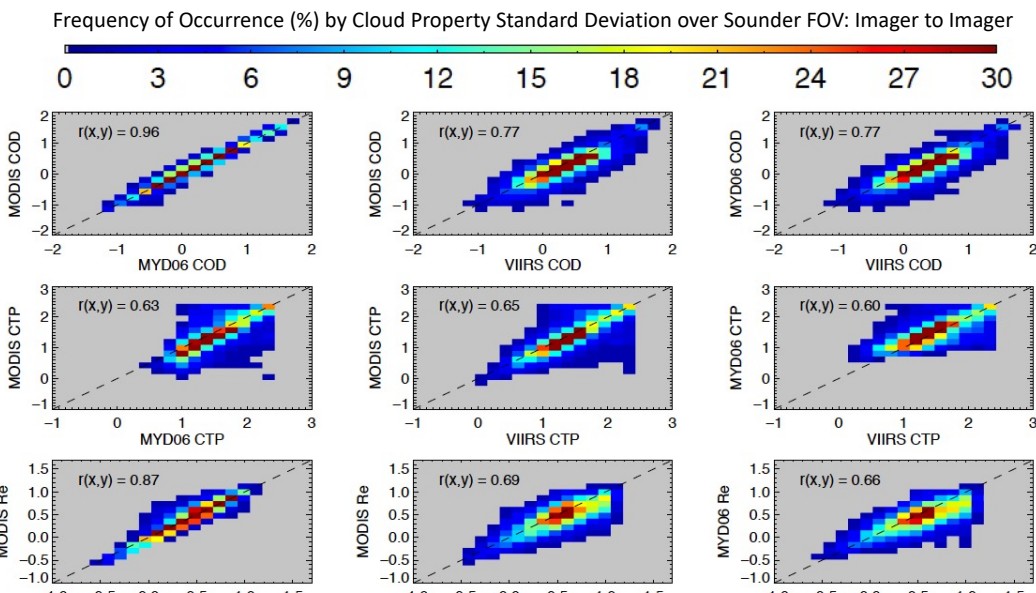

Figure 8. Similar to Fig. 7, except showing comparisons of standard deviation of cloud properties
over the sounder FOV, which are calculated using the finer resolution imager observations
collocated with the same sounder FOV. All the results are presented on log10 scale. Linear
correlation coefficients between the variables on x- and y- axes are given in each plot.



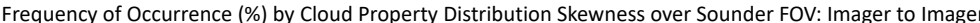

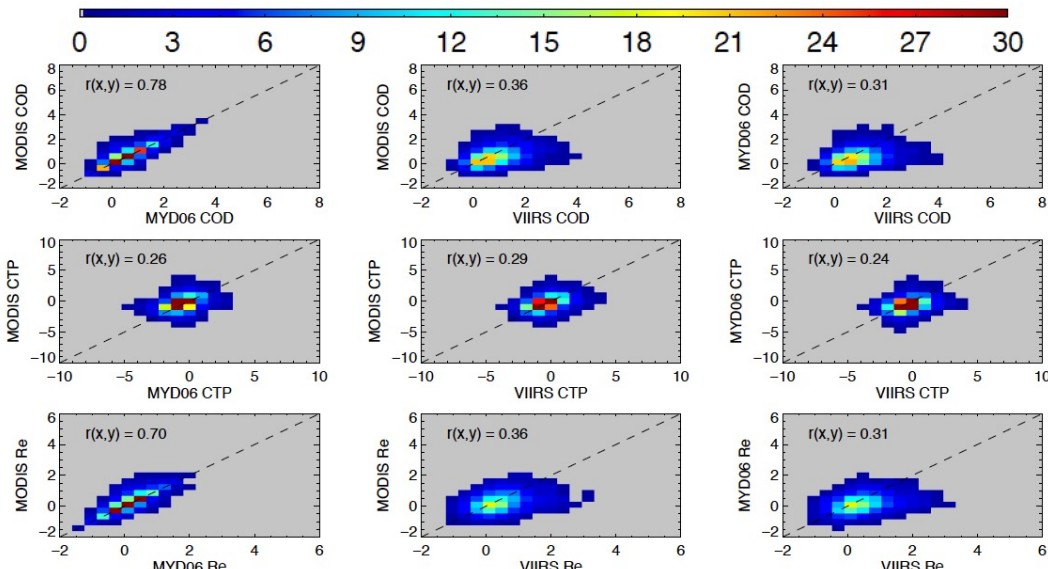

Figure 9. Similar to Figs. 8 and 7, except cloud property skewness over sounder FOV is used in
the comparison. Results are shown in linear scale. Linear correlation coefficients between the
variables on x- and y-axes are given in each plot.

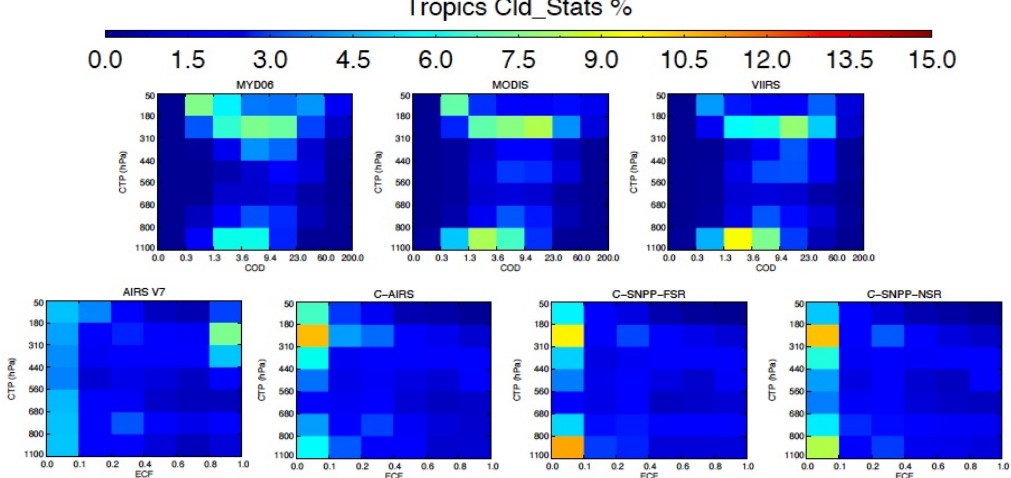

Figure 10. The 2-dimensional histograms calculated using SNOs from the focus days in the
tropics (30°N~30°S). The top row shows results for MODIS and VIIRS, for which the ISCCP
type of COD-CTP joint histograms are presented by summarizing the histograms over individual
AIRS and CrIS FOV. Note that no averaging over sounder FOV is taken for this comparison.
From left to right show results of MYD06, *Aqua*-MODIS continuity, and *SNPP*-VIIRS
continuity cloud products. The bottom row shows results for AIRS and CrIS, and joint
distributions are calculated on the imager effective CTP and ECF space. From left to right, data
from AIRS Version 7 (AIRS V7), CLIMCAPS-*Aqua* (C-AIRS), CLIMCAPS-*SNPP* FSR (C-
*SNPP*-FSR), and CLIMCAPS-*SNPP* NSR (C-*SNPP*-NSR) are used in the calculation.



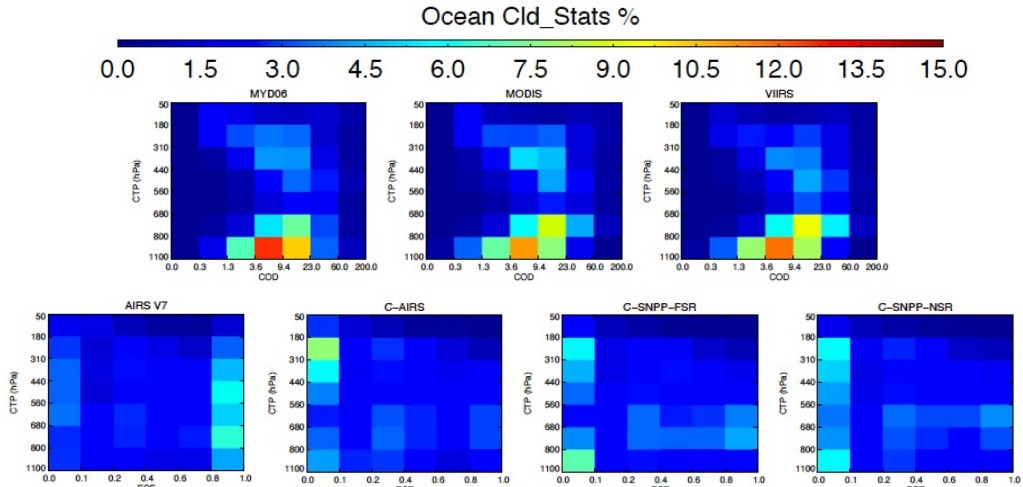

Figure 11. Similar to Fig. 10, except showing results calculated using data over 60°N~60°S
ocean. Sounder land fraction < 0.1 is used to determine ocean surfaces.



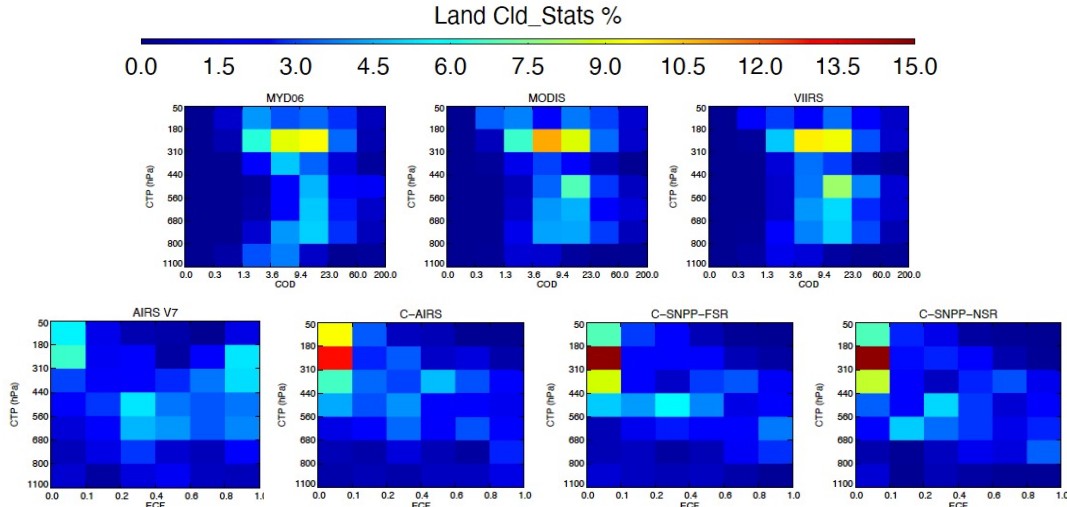

Figure 12. Similar to Figs. 11 and 10, except showing results calculated using data over 60°N~60°S land. Sounder land fraction > 0.9 is used to determine land surfaces.

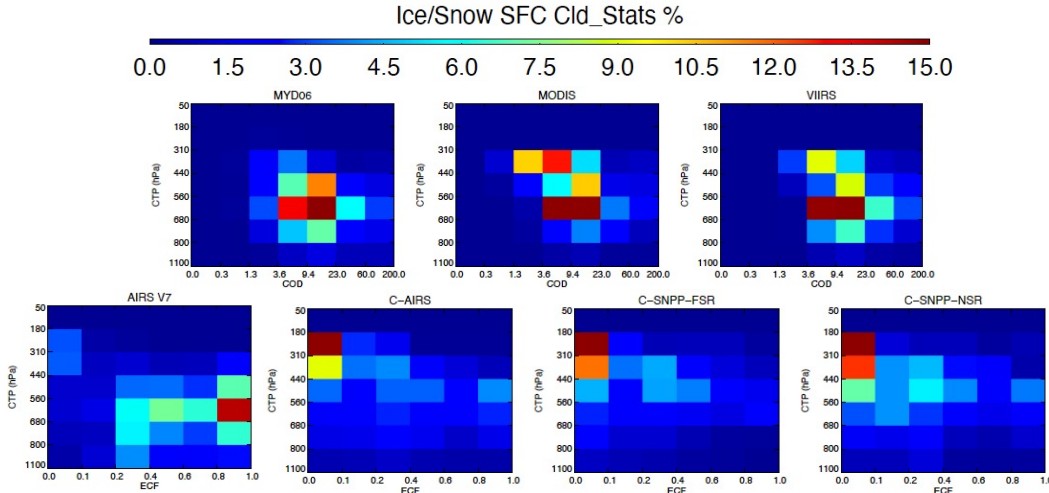

Figure 13. Similar to Fig. 10-12, except showing results calculated using data over snow and ice
covered surfaces. Sounder retrieved surface classes are used to identify cases.

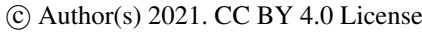


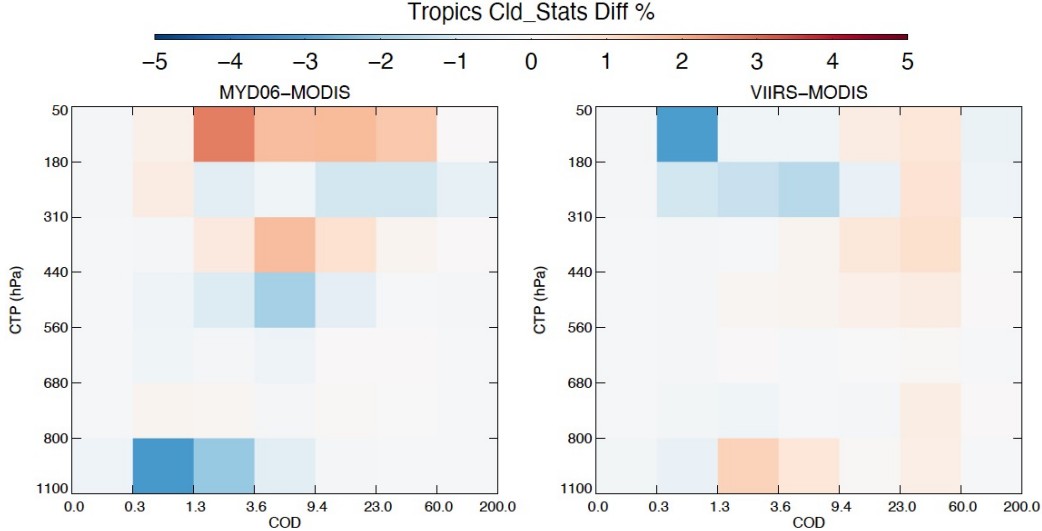

Figure 14. Differences of the imager CTP-COD cloud histograms in the tropics: between the
MYD06 and *Aqua*-MODIS continuity products (left), and between the *Aqua*-MODIS and *SNPP*-
VIIRS continuity cloud products (right).



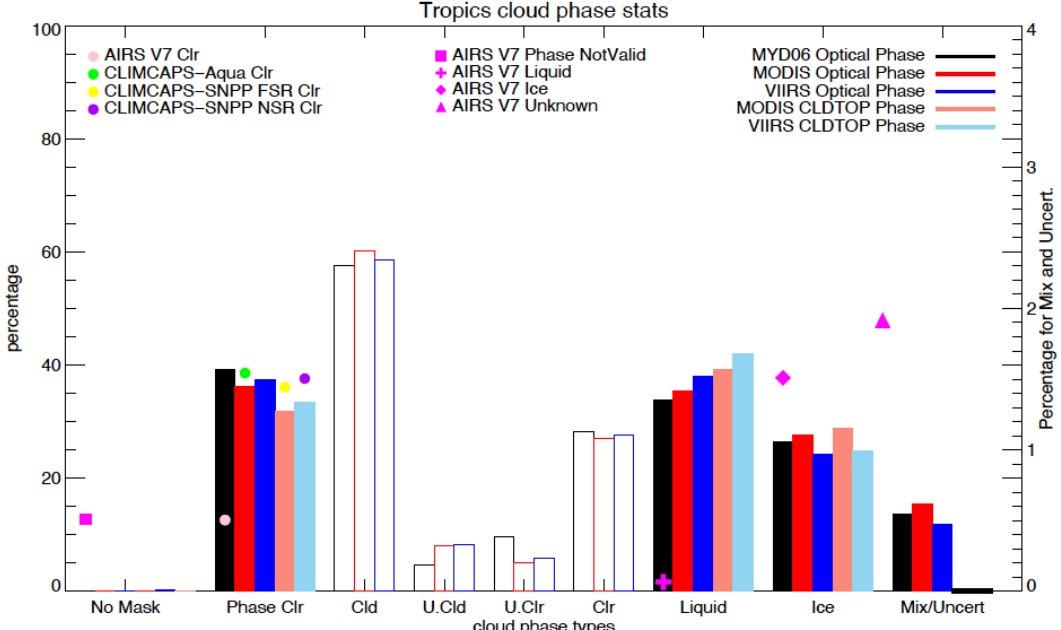

Figure 15. The histograms of cloud thermodynamic phases (solid color bars) and cloud mask
(hollow color bars) in the tropics (30°N~30°S) from the imager cloud products calculated using
retrievals on SNOs from the seven focus days. The frequency of clear sky detected by IR
sounders using thresholds of ECF < 0.01 is also shown by colored solid circles. AIRS Version 7
cloud thermodynamic phase is shown by magenta symbols. Color of the bars corresponds with
different imager cloud retrievals for cloud mask and cloud thermodynamic phase determined in
the optical property retrieval (Cloud_Phase_Optical_Properties): black for MYD06, red for *Aqua*
MODIS continuity products (CLDPROP_MODIS), and blue for *SNPP* VIIRS continuity
products (CLDPROP_VIIRS), respectively. Cloud_Phase_Optical_Propertes reports flags
indicating cloud mask not determined for pixel (no mask), clear sky (Phase Clr), liquid water
cloud (Liquid), ice cloud (ICE), or undetermined phase (Mix/Uncert). Cloud phases reported by
Cloud_Phase_Cloud_Top_Properties in the MODIS-VIIRS continuity cloud products are also
evaluated and results are shown with pink (MODIS) and light blue (VIIRS) bars, which shows
flags indicating cloud free (Phase Clr), water cloud (Liquid), ice cloud (ICE), mixed phase cloud
or undetermined phase (Mix/Uncert). Note that the Mix/Uncert phase category for imager
products is shown with the y-axis on the right due to its much smaller frequency of occurrence.
Cloud mask histograms of Not determined (No Mask), Cloudy (Cld), Uncertain (U. Cld),
Probably Clear (U. Clr), and Confident Clear (Clr) are shown in the figure following this color
convention but using hollow bars. For IR sounder clear sky frequency, results from AIRS V7
(pink), CLIMCAPS-AIRS (green), CLIMCAPS-*SNPP* FSR (yellow), and CLIMCAPS-*SNPP*
NSR (purple) are overlaid on top of the Phase Clr histograms for sounder-imager clear sky
detection comparison.





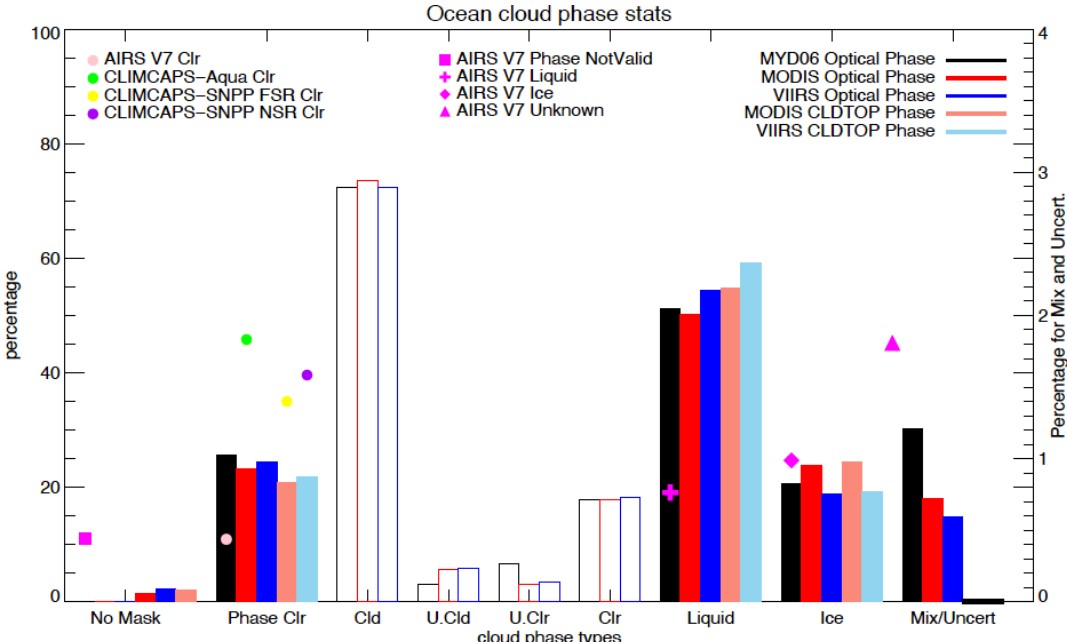

Figure 16. Similar to Fig. 15, except showing results calculated using data over 60°N~60°S ocean. Sounder land fraction < 0.1 is used to determine ocean surfaces.





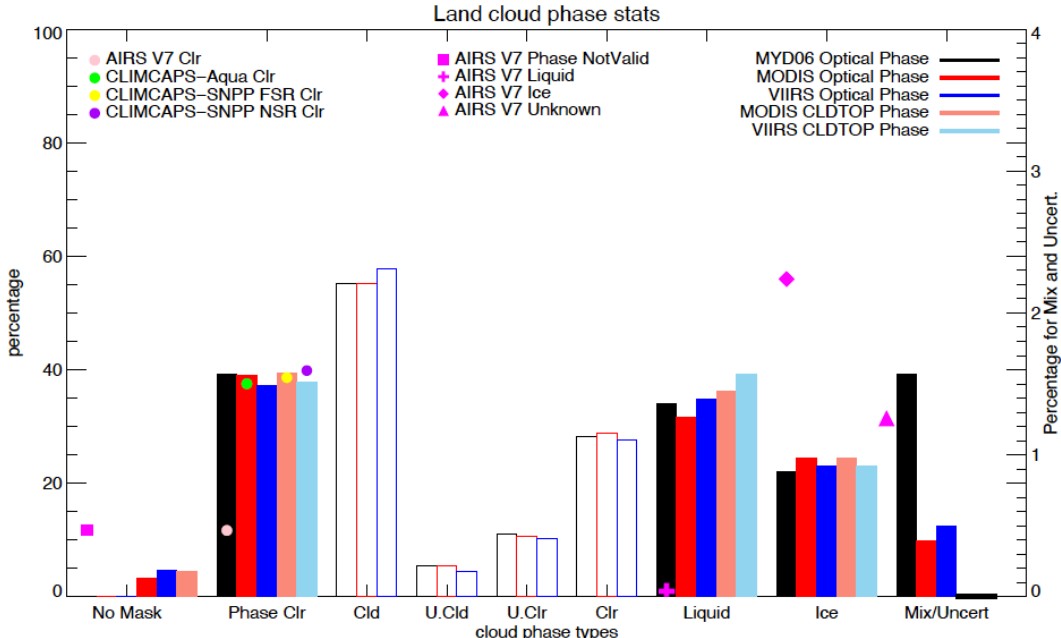

Figure 17. Similar to Figs. 16 and 15, except showing results calculated using data over 60°N~60°S land. Sounder land fraction > 0.9 is used to determine land surfaces.





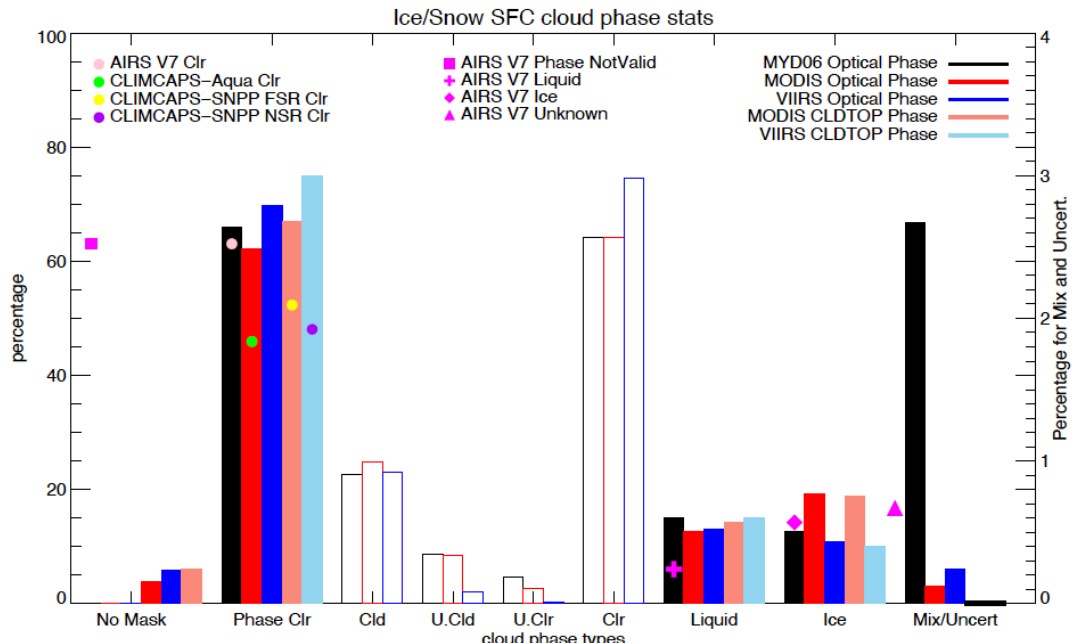

Figure 18. Similar to Figs. 15-17, except showing results calculated using data over snow and ice
covered surfaces. Sounder retrieved surface classes are used to identify cases.