# Peer review of "Evaluating the Consistency and Continuity of Pixel-Scale Cloud 1"

_Atmospheric Measurement Techniques, 2021_

## Author Comment (AC1)

[Figure]

Figure S1. Similar to Fig. 3, but for ice-cloud-only sounder FOVs, which are determined as more than 80% of collocated MYD06 cloudy pixels reported as ice phase clouds in the optical property cloud phase retrieval. Comparisons of ECF (top row) and effective CTP (bottom row) derived from different sounder retrieval algorithms. Linear correlation coefficients are calculated for cloud properties obtained from retrieval products indicated on the axes and are given on top of the each plot. From left to right, results comparing AIRS Version 7 with CLIMCAPS-*Aqua* (C-A), CLIMCAPS-*SNPP* FSR (C-S-F), and CLIMCAPS-*SNPP* NSR (C-S-N) are shown using joint distributions of frequency of occurrence (%). The data points located in regions poleward of 60° are excluded. Cases are included only when both retrievals in comparison (x- and y-axes of the plot) report valid retrievals.

[Figure]

Figure S2. Similar to Fig. S1, but for liquid-cloud-only sounder FOVs, which are determined as more than 80% of collocated MYD06 cloudy pixels reported as liquid phase clouds in the optical property cloud phase retrieval.

[Figure]

Figure S3. Similar to Fig. 5, but for ice-cloud-only sounder FOVs, which are determined as more than 80% of collocated MYD06 cloudy pixels reported as ice phase clouds in the optical property cloud phase retrieval. Comparisons of sounder and imager derived cloud properties shown by joint distribution of case frequency of occurrence. Top row shows evaluation of sounder-derived ECF by cloud optical depth (COD, in log10 scale) from the MYD06 products. The middle row compares the sounder effective CTP with CTP from MYD06 overlaid by the magenta contours showing the mean ECF from the corresponding sounder retrievals. The bottom row is similar to the middle row except that the cases with sounder ECF < 0.1 are removed from the comparison. Different sounder retrieval algorithms are included. From left to right, data from AIRS Version 7, CLIMCAPS-*Aqua* (C-A), and CLIMCAPS-*SNPP* FSR (C-S-F) are used. The data points located in regions poleward of 60° are excluded. Cases are included only when both retrievals in comparison (x- and y-axes of the plot) report valid retrievals. The cloud properties from MODIS pixels collocated within the same sounder FOV are averaged before comparing with the IR sounder data. Linear correlation coefficients between the variables on x- and y-axes for different conditions are given in each plot.

[Figure]

Figure S4. Similar to Fig. S3, except using the MODIS continuity cloud product (CLDPROP_MODIS).

[Figure]

Figure S5. Similar to Fig. S3, but for liquid-cloud-only sounder FOVs, which are determined as more than 80% of collocated MYD06 cloudy pixels reported as liquid phase clouds in the optical property cloud phase retrieval.

[Figure]

Figure S6. Similar to Fig. S5, except using the MODIS continuity cloud product (CLDPROP_MODIS).

[Figure]

Figure S7. Similar to Fig. 7, but for ice-cloud-only sounder FOVs, which are determined as more than 80% of collocated MYD06 cloudy pixels reported as ice phase clouds in the optical property cloud phase retrieval. Comparison of cloud optical depth (COD, in log10 scale), cloud top pressure (CTP, hPa), and effective particle size (Re, μm) retrieved by MODIS and VIIRS cloud algorithms. The mean imager cloud properties over corresponding sounder FOVs are compared over the SNOs. From left to right show the results of following comparisons: *Aqua* MODIS continuity cloud products (CLDPROP_MODIS) with MYD06, CLDPROP_MODIS with *SNPP*-VIIRS continuity cloud products (CLDPROP_VIIRS), and MYD06 with CLDPROP_VIIRS, respectively. Linear correlation coefficients between the variables on x- and y-axes are given in each plot.

[Figure]

Figure S8. Similar to Fig. S7, but for liquid-cloud-only sounder FOVs, which are determined as more than 80% of collocated MYD06 cloudy pixels reported as liquid phase clouds in the optical property cloud phase retrieval.

[Figure]

Figure S9. Similar to Fig. S7, except showing comparisons of standard deviation of cloud properties over the ice-cloud-only sounder FOV, which are calculated using the finer resolution imager observations collocated with the same sounder FOV. All the results are presented on log10 scale. Linear correlation coefficients between the variables on x- and y- axes are given in each plot.

[Figure]

Figure S10. Similar to Fig. S9, but for liquid-cloud-only sounder FOVs, which are determined as more than 80% of collocated MYD06 cloudy pixels reported as liquid phase clouds in the optical property cloud phase retrieval.

---

## Author Response (AR1)

**Response to RC1:**

We would like to thank the reviewer for the encouragement on this study and the suggestions on our manuscript. Below we have responded point by point to all the comments. Line numbers and page numbers are based on the revised version (clean version). **The reviewer's original comments are in bold** and the revised text is shown within quotation marks.

**1. Line 46, "The 2017 US National Academy Decadal Survey (ESAS 2017)", should this be 2018 instead of the 2017 Decadal Survey?**

We followed the example given by https://www.nationalacademies.org/our-work-decadal-survey-for-earth-science-and-applications-from-space. The 2017-2027 Decadal Survey for Earth Science and Applications from Space is referred to as ESAS 2017. The report "Thriving on Our Changing Planet: A Decadal Strategy for Earth Observation from Space" was published in 2018.

- 2. Line 183, please specify what exactly version of ECMWF data was used here. We revised the description on the ECMWF data used in the AIRS V7 SCCNN and added version information on Line 188-191: "trained using a few months of AIRS/AMSU radiances and European Center for Medium-Range Weather Forecasting (ECMWF) Integrated Forecast System (IFS) 3-hourly forecast fields that are collocated to AIRS (including updates since Version CY31R1: https://www.ecmwf.int/en/forecasts/documentation-and-support/changes-ecmwfmodel) (Milstein and Blackwell 2016)."
- 3. Line 191, Hook (2019) was cited here for the CAMEL surface emissivity. This might be OK. But the CAMEL has two formal publications, https://doi.org/10.3390/rs10040643 and https://doi.org/10.3390/rs10050664. The authors might consider these two papers as references here. Thank you. We have added the two publications to our reference for CAMEL surface emissivity.
- **4.** In the legend of Figure 2, "MODIC Con." Should be "MODIS Con.". Thank you. The typo has been corrected.
- 5. In figures 15-18, the arrangements of the bar plot are a bit confusing. For two groups, "No Mask" and "Mix/Uncert", are bars arranged in the same order as in other groups? It looks not like the case on my screen. Thank you for catching this error. We found that wrong colors were used in the previous version. This error has been corrected in fig. 15-18.
- 6. There are five places that an "i" is missing in "Cloud\_Phase\_Optical\_Propertes". Thank you. The typo has been corrected.

**Response to RC2:**

We would like to thank the reviewer for the encouragement and suggestions on this study. Below we have responded point by point to all the comments. Line numbers and page numbers are based on the revised version (clean version). **The reviewer's original comments are in bold** and the revised text is shown within quotation marks.

**Main Comments:**

1. The algorithm descriptions could have a few more details, especially for the imagers. For example, the different shortwave infrared channels used by MODIS and VIIRS have 'implications' as mentioned in line 218, but I suggest to briefly explain what they are for liquid and ice cloud effective radii retrievals. Also, some brief description on the "differences in LUTs" that are mentioned on line 241 would be good. It is also unclear to me what the difference between CLDPROP and MYD06 ice phase algorithm is. It is stated on line 241 that CLDPROP "removes the dependence on the cloud top solution method in MYD06." Do you mean it does not rely on cloud top height?

Following the reviewer suggestions, we have added more discussions to summarize the major differences between the imager retrieval algorithms, which include the SWIR channel differences and the impact on cloud microphysical property retrieval, and explanation on LUT updates, and the IR channel differences and impact on cloud top determination, as well as more details on cloud phase algorithm changes. Please see Line 240-263 on Page 10-11: "The continuity CLDPROP products use only spectral channels common to both MODIS and VIIRS. The algorithm has direct heritage with the Collection 6.1 MODIS atmosphere cloud retrievals (MYD06), with cloud-top property datasets provided by the CLouds from AVHRR (the Advanced Very High Resolution Radiometer) - Extended (CLAVR-x) processing system (Heidinger et al. 2012, 2014) to account for more limited information for cloud-top property retrieval. CLAVR-x produces cloud phase reported as Cloud Phase Cloud Top Properties in the MODIS-VIIRS continuity cloud products. Since VIIRS does not have IR channels in the 13 µm CO2 absorption band, the MODIS CO2 slicing solution for cloud top pressure retrievals for cold clouds is replaced with an IR window channel optimal estimation approach coupled with a Cloud-Aerosol Lidar and Infrared Pathfinder Satellite Observations (CALIPSO)derived a priori (Heidinger et al. 2019). This in turn affects the optical property cloud phase algorithm (reported as Cloud Phase Optical Properties in CLDPROP products), which removes the cold cloud sanity check applied in the MOD06/MYD06 that is based on the CO2-slicing solution. The spectral mismatch of the MODIS 2.13 µm and VIIRS 2.25 µm channels also bring further changes to the Cloud Phase Optical Properties retrieval by modifying the spectral cloud effective radius (Re) test approach. In the Version 1.1 MODIS-VIIRS continuity cloud product used in this study, the 2.25 µm test is omitted and the 1.61 µm test is duplicated. Moreover, this channel spectral differences compel changes in the look-up tables (LUT) of spectral liquid cloud reflectance used in the retrieval, which include the use of an updated liquid water imaginary index of refraction dataset in the shortwave infrared region (Kuo et al. 1993) and an updated complex index of refraction dataset for 3.7 µm (Wagner et al. 2005).

Such differences in LUTs result in changes of cloud effective particle size (Re) (Platnick et al. (2020) that, along with cloud optical depth (COD), are used to derive cloud water path. Moreover, the ice crystal absorption at 2.25  $\mu$ m is weaker than that at 2.13  $\mu$ m.".

2. I would suggest to separate ice-only and liquid-only FOVs in figures 7, 8 and 9. Alternatively, these could be provided in a supplement and the differences in results for ice-only and liquid-only FOVs can be discussed in the paper. Especially ice crystal absorption is much weaker at 2.25 micron compared to 2.13 micron, so there could be greater differences for ice clouds between MODIS and VIIRS than for liquid clouds.

The comparisons on ice- and liquid-cloud-only FOVs are carried out following reviewer's suggestions. Results are included in the supplement file and main differences are discussed in the manuscript.

- On sounder-sounder comparison, please see Fig. S1-S2 and discussions on Line 404-409: "Further separating the sounder FOVs into ice- and liquid-cloud-only categories shows that such inconsistency in cloud amount detection between the sounder algorithms exist in both categories as illustrated in Fig. S1 and S2. The sounder FOV is determined as ice/liquid-cloud-only when over 80% of collocated cloudy MODIS pixels are in ice/liquid thermodynamic phase in the MYD06 optical property cloud phase retrievals. Better agreements between sounder cloud products are found for ice-cloud-only FOVs.".
- On sounder-imager comparison, please see Fig. S3-S6 and discussions on Line 458-462: "The results are further analyzed for ice- and liquid-cloud-only sounder FOVs (Fig. S3-S6), which are determined using the same criteria as in the previous section. It is clear that the disagreements between the sounder and imager CTP retrievals are mainly originated from the liquid-cloud-only sounder FOVs (Fig. S5 and S6), while good agreements are found for ice-cloud-only conditions (Fig. S3 and S4)."
- On imager-imager comparison, please see Fig. S7-S10 and discussions on Line 475-477: "especially for cold clouds as shown in Fig. S7, where the correlation coefficients for CTPs from different imager cloud retrievals are less than 0.52 for icecloud-only conditions (Fig. S7) but larger than 0.79 for liquid-cloud-only cases (Fig. S8).". Line 483-491: "Separating results into ice- and liquid-cloud-only conditions, the COD (Re) correlation coefficients between the MODIS and VIIRS continuity cloud products are 0.84 (0.70) and 0.82 (0.75) for ice- and liquid-cloud-only conditions, respectively, as shown in Fig. S7 and S8. Although such good agreements between the two imagers are encouraging, the correlation for Re from the two CLDPROP products is lower than that for COD, with a much weaker correlation on the ice cloud Re retrievals. This reflects the effect of spectral channel and spatial resolution differences between MODIS and VIIRS, as well as the related adjustments made to the continuity algorithms, such as the liquid phase LUT for cloud microphysical retrievals, especially the impact of weaker ice crystal absorption at 2.25 µm (VIIRS) than at 2.13 µm (MODIS)." Line 501-503: "with a much lower correlation on CTP (r = 0.44) for ice-cloud-only conditions (Fig. S9) but a high correlation (r = 0.71) for liquidcloud-only FOVs (Fig. S10)."Line 504-506: "The impact from the differences in CTP

algorithms thus shows up more strongly on the higher statistical moments and on cold cloud scenes.".

3. Related to this, I wonder if there also are differences between ice-only and liquid-only FOVs in the sounder to sounder and the sounder to imager comparisons. Could you at least comment on that?

Please see our response to Main Comment #2.

**Minor and specific comments:**

- 1. Line 114: I suggest to include an outline of the paper as is customary. The outline is added on Line 113-118 as suggested: "This article is organized as follows. Section 2 describes various cloud products and their retrieval algorithms analyzed in this study, as well as the method used to create pixel-scale collocated datasets between sounders and imagers across different satellites. Section 3 shows the detailed comparisons of cloud properties and their joint histograms from different algorithms and sensors, and the discussions on implications on retrieval algorithm development and instrument differences. A summary and set of conclusions are presented in Section 4."
- **2.** In Table 1, I'd suggest to include the spatial resolution of the products The spatial resolution is added in Table 1.
- 3. In table 1, I suggest to spell out NSR and FSR, so it's clear what the difference is between those two rows.

NSR and FSR are spelt out in Table 1 as suggested.

4. Figure 1 and 2: The yellow lines are very hard to see, especially the dashed one. I suggest to use a different color.

Yellow lines in these figures and symbols in other plots are changed to a different color (dark goldenrod) for better visualization of the results.

5. Figures 5 and 6: The addition of the magenta lines make the middle panel plots very busy, and the number indicating the contour line values are almost impossible to read. These should be made more clear. Making these panels bigger might help.

Fonts in all plots are increased as suggested. The number of contour levels in Fig. 5 and 6 is reduced so that the 0.1 ECF contour is more clear, with the third rows in these plots to show results removing the ECF < 0.1 FOVs.